# Enantiomeric glycosylated cationic block co-beta-peptides eradicate *Staphylococcus aureus* biofilms and antibiotic-tolerant persisters

Kaixi Zhang[1,2,18], Yu Du [1,2,17,18], Zhangyong Si[1,2,18], Yang Liu[2,3,18], Michelle E. Turvey[4], Cheerlavancha Raju[1,2], Damien Keogh[1,2], Lin Ruan[1,2], Subramanion L. Jothy[1,2], Sheethal Reghu[1,2], Kalisvar Marimuthu[5,6], Partha Pratim De[7], Oon Tek Ng[5,6,8], José R. Mediavilla[9], Barry N. Kreiswirth[9], Yonggui Robin Chi[10], Jinghua Ren[11], Kam C. Tam[12], Xue-Wei Liu[2,10], Hongwei Duan [1,2], Yabin Zhu[13], Yuguang Mu [3], Paula T. Hammond [14,15], Guillermo C. Bazan [16], Kevin Pethe[2,3,8]* & Mary B. Chan-Park [1,2,8]*

The treatment of bacterial infections is hindered by the presence of biofilms and metabolically inactive persisters. Here, we report the synthesis of an enantiomeric block co-beta-peptide, poly(amido-D-glucose)-*block*-poly(beta-L-lysine), with high yield and purity by one-shot one-pot anionic-ring opening (co)polymerization. The co-beta-peptide is bactericidal against methicillin-resistant *Staphylococcus aureus* (MRSA), including replicating, biofilm and persister bacterial cells, and also disperses biofilm biomass. It is active towards community-acquired and hospital-associated MRSA strains which are resistant to multiple drugs including vancomycin and daptomycin. Its antibacterial activity is superior to that of vancomycin in MRSA mouse and human ex vivo skin infection models, with no acute in vivo toxicity in repeated dosing in mice at above therapeutic levels. The copolymer displays bacteria-activated surfactant-like properties, resulting from contact with the bacterial envelope. Our results indicate that this class of non-toxic molecule, effective against different bacterial sub-populations, has promising potential for the treatment of S. *aureus* infections.

---

[1] School of Chemical and Biomedical Engineering, Nanyang Technological University, 62 Nanyang Drive, Singapore 637459, Singapore. [2] Centre for Antimicrobial Bioengineering, Nanyang Technological University, 62 Nanyang Drive, Singapore 637459, Singapore. [3] School of Biological Sciences, Nanyang Technological University, 60 Nanyang Drive, Singapore 637551, Singapore. [4] Infectious Disease Interdisciplinary Research Group, Singapore-MIT Alliance for Research & Technology Centre, 1 Create Way, Singapore 138602, Singapore. [5] Department of Infectious Diseases, Tan Tock Seng Hospital, 11 Jalan Tan Tock Seng, Singapore 308433, Singapore. [6] National Centre for Infectious Diseases, 16 Jalan Tan Tock Seng, Singapore 308442, Singapore. [7] Department of Laboratory Medicine, Tan Tock Seng Hospital, 11 Jalan Tan Tock Seng, Singapore 308433, Singapore. [8] Lee Kong Chian School of Medicine, Nanyang Technological University, 59 Nanyang Drive, Singapore 636921, Singapore. [9] Center for Discovery and Innovation, Hackensack Meridian Health, Nutley, NJ 07110, USA. [10] Division of Chemistry & Biological Chemistry, School of Physical & Mathematical Sciences, Nanyang Technological University, 21 Nanyang Link, Singapore 637371, Singapore. [11] Cancer Center, Union Hospital, Huazhong University of Science & Technology, Wuhan 430022 Hubei, China. [12] Department of Chemical Engineering, Waterloo Institute for Nanotechnology, University of Waterloo, Ontario N2L 3G1, Canada. [13] Medical School of Ningbo University, Ningbo, 315211 Zhejiang, China. [14] Koch Institute for Integrative Cancer Research, Massachusetts Institute of Technology, Cambridge, MA 02139, USA. [15] Department of Chemical Engineering, Massachusetts Institute of Technology, Cambridge, MA 02139, USA. [16] Department of Chemistry and Biochemistry, University of California Santa Barbara, Santa Barbara, CA 93106-9510, USA. [17]Present address: Fujian Institute of Research on the Structure of Matter, Chinese Academy of Sciences, 155 Yangqiao Road West, 350002 Fuzhou, China. [18]These authors contributed equally: Kaixi Zhang, Yu Du, Zhangyong Si, Yang Liu. *email: kevin.pethe@ntu.edu.sg; mbechan@ntu.edu.sg

Antimicrobial resistance in bacteria is a serious and growing clinical problem, eroding the therapeutic armamentarium and leaving limited treatment options for certain infections. Compounding the difficulty of treating antibiotic-resistant strains is the presence of persisters, subpopulations that are antibiotic-tolerant due to metabolic inactivity[1,2], and the capacity of bacteria to develop biofilms[3], both of which lead to chronic and recurrent infections[4–7]. The World Health Organization (WHO) recently published a priority list of bacteria for which new antibiotics are urgently needed[8]. Methicillin-resistant *Staphylococcus aureus* (MRSA), a WHO high-priority pathogen, is a leading cause of mortality due to antibiotic-resistant infections[9,10]. Initially restricted to hospitals and healthcare settings, MRSA is causing an increasing number of infections in the community[11,12]. MRSA is associated with poor clinical outcomes[13]: it causes frequent skin and soft tissue infections[14] and can disseminate, resulting in life-threatening bloodstream infections, endocarditis, bone and joint infections, as well as pneumonia[15,16]. *S. aureus* is prone to form biofilms and also exists in the form of metabolically inactive antibiotic-tolerant persister phenotype[3]. Last-resort antibiotics such as vancomycin are largely ineffective against *S. aureus* persisters and biofilms[17]. New therapeutics are needed to combat the spread of difficult to treat drug-resistant *S. aureus* infections. Alternative antibacterial agents should have bactericidal activity against replicating cells, persisters, and established biofilms. Cationic alpha-peptides and membrane-active agents have been investigated as alternative antimicrobials to combat biofilms and persisters[18,19], but unselective toxicity is a complicating factor[20].

Amongst the various synthetic polymer families being explored as peptidomimetics[21–25], beta-peptides are promising because they can exhibit biological activity comparable to natural peptides, but have better proteolytic stability[26], and are usually amphiphilic and non-mutagenic[27]. Beta-peptides have been considered for use in diverse therapeutic applications such as antimicrobial agents[28–30], vaccine drugs[31], protein–protein interaction inhibitors[32,33], and drug delivery[34,35]. Alpha-peptide antimicrobials are known to form facially amphiphilic (FA) structures that enhance the bactericidal properties but tend to be hemolytic and toxic[36]. Compared to alpha-peptides, beta-peptides have an extra methylene group in the backbone. The hydrophobicity of beta-peptides may be tuned by the structure of the side chains. Further, beta-peptides may be designed to form foldamers exhibiting diverse secondary structures, such as helices and beta-sheets[37–39] and complex tertiary and quaternary structures[40].

Munoz-Guerra and colleagues reported the first research on nylon-3 and analogs, which included the synthesis and helical propensity of these beta-peptides[41–43]. In the development of antimicrobial beta-peptides, previous efforts focus mainly on random co-beta-peptides and optimization of their cationic versus hydrophobic beta-lactam residues to reduce hemolysis whilst maintaining a good bactericidal effect[44–47]. There is no reported work on glycosylated block co-beta-peptides. Block co-poly(beta-peptides) are interesting as they may show unique combinations of properties displayed by the individual blocks, which are as yet under-exploited for the development of next-generation antibacterials. Also, a strategy for the facile synthesis of block co-beta-peptides has not been previously reported.

In this study, we report a simple one-shot one-pot anionic ring opening (co)polymerization (AROP) strategy to synthesize a series of enantiomeric block co-beta-peptides, which cannot be made by sequential copolymerization. Two beta-lactam monomers with contrasting reactivities—a protected D-glucose (**DGu$_p$**) beta-lactam and a protected cationic beta-L-lysine (**BLK$_p$**) beta-lactam—can be block copolymerized in one shot. The resulting optimized block co-beta-peptide, **PDGu(7)-*block*-PBLK(13)**, is non-cytotoxic and non-hemolytic in vitro. Further, the block co-beta-peptide has interesting biological properties. Unlike classical antibiotics, **PDGu(7)-*b*-PBLK(13)** retains potency against MRSA persister cells and biofilms. It is active against both the community-acquired (CA-) and hospital-associated (HA-) MRSA strains. The block copolymer also effectively removes biofilm biomass but the homocationic beta-peptide (**PBLK(20)**) cannot. The block copolymer is bactericidal against MRSA in various murine models of systemic acute and established infections, and also in an ex vivo human skin infection model, while having no in vivo acute toxicity in murine repeated dosing studies. This study opens up possibilities of treatment for recalcitrant MRSA infections.

## Results

**Synthesis of the (co)polymers via one-shot one-pot AROP.** The monomers *N*-Cbz-*β*-lactam-L-lysine (**BLK$_p$**) and *O*-Bn-*β*-lactam-D-glucose (**DGu$_p$**) were synthesized and verified by nuclear magnetic resonance (NMR) spectroscopy (Supplementary Methods, Supplementary Figs. 1–3). The copolymer synthetic strategy relies on the observation that the homopolymerization of *N*-Cbz-*β*-lactam-L-lysine monomer (**BLK$_p$**) is much slower than the homopolymerization of *O*-Bn-*β*-lactam-D-glucose monomer (**DGu$_p$**) (Supplementary Table 1). When **DGu$_p$** and **BLK$_p$** monomers (10:10, mole/mole) were mixed together in tetrahydrofuran, **DGu$_p$** was totally consumed in 8 min while the **BLK$_p$** monomer required 8 h for complete reaction (Fig. 1a–c, Supplementary Figs. 4, 5). The molecular weight of the product increased linearly over an 8-h period during which **DGu$_p$** disappeared rapidly in the first few minutes, while **BLK$_p$** was consumed gradually over the next few hours (Fig. 1d). A plot of molecular weight ($M_n$) versus **BLK$_p$** conversion (Fig. 1e) shows a linear relationship and the Đ values of the products remain small (1.06–1.12). These results are consistent with the growth of a single copolymer chain through rapid consumption of **DGu$_p$** followed by slower but contiguous incorporation of **BLK$_p$**, and thus provide evidence for a 'block-like' structure of the resulting copolymer. A series of poly(Bn-amido-D-glucose)-*block*-poly(Cbz-beta-L-lysine) (**PDGu$_p$(x)-*b*-PBLK$_p$(y)**) block copolymers, with varying ratios of *x* to *y* but constant target total degree of polymerization of 20, i.e. $(x + y) = 20$, was synthesized (Fig. 1a, Supplementary Fig. 6). The molecular weights are close to the design values based on gel permeation chromatography (GPC) relative to polystyrene standards, confirming that the AROP process is well-controlled (Fig. 1f, Supplementary Table 2).

After one-step deprotection, the final products **PDGu(x)-*b*-PBLK(y)** were obtained with overall yields greater than 65% (Supplementary Fig. 7). NMR spectroscopy measurements of **PDGu(x)-*b*-PBLK(y)** show two sets of signals belonging to **PDGu** and **PBLK**, respectively, corroborating their block rather than random structures (Supplementary Figs. 8–22). NMR spectra also show that the ratios of **DGu** to **BLK** in **PDGu(x)-*b*-PBLK(y)** after purification deviate slightly from the stoichiometric ratios of added monomers. For example, the actual composition of **DGu** and **BLK** in **PDGu(10)-*b*-PBLK(10)** is **PDGu(7)-*b*-PBLK(13)**; the **PBLK** block is 66 mol% versus the design value of 50 mol%. This trend is repeatable and can be seen in other compositions (Table 1). The molecular weights ($M_n$) of the homocationic **PBLK(20)**, homosugar **PDGu(20)**, and copolymer **PDGu(7)-*b*-PBLK(13)** were, respectively, 3012 Da, 3159 Da, and 3391 Da, as measured by the Matrix Assisted Laser Desorption/Ionization-Time of Flight (MALDI-TOF) mass spectroscopy (Supplementary Fig. 23).

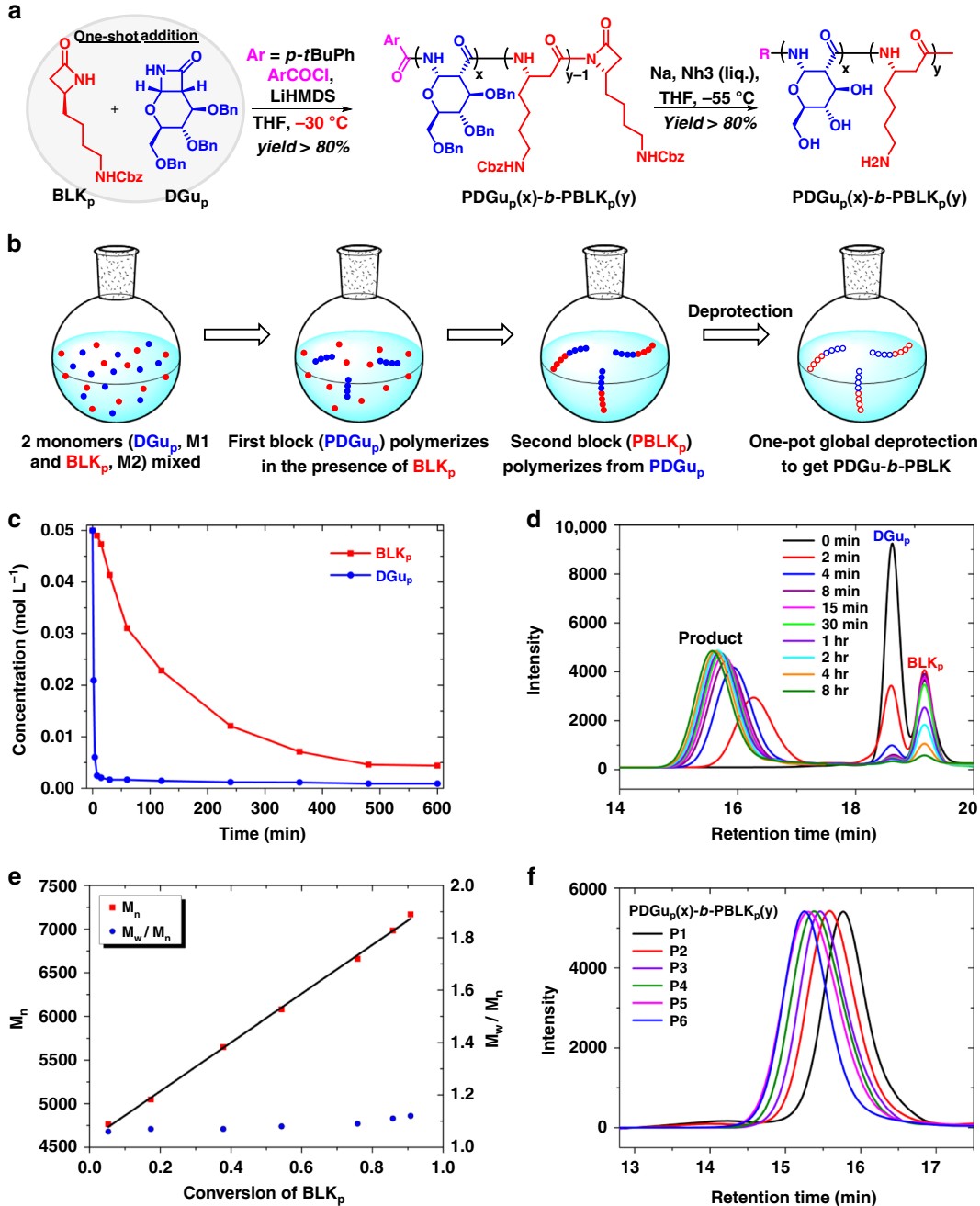

**Fig. 1** Facile one-shot one-pot synthesis of **PDGu(x)-b-PBLK(y)** block copolymer. **a** Synthetic scheme of **PDGu(x)-b-PBLK(y)**. **b** One-shot addition of both monomers (**DGu_p** and **BLK_p**) leads to block copolymerization when the monomers have contrasting reactivities. **c–e** Kinetic studies and **f** GPC measurements verify the well-controlled single chain block architecture of **PDGu_p(x)-b-PBLK_p(y)**. **c** Remaining monomer concentration vs time. **d** GPC curves of partially polymerized products at selected quenching times. **e** Molecular weight ($M_n$) and molecular weight distribution (Đ) as a function of conversion of **BLK_p**. **f** GPC of protected-(co)polymers

When we attempted the synthesis of **PDGu_p(10)-b-PBLK_p(10)** by the sequential addition of **DGu_p** (M1) followed by **BLK_p** (M2) at −30 °C (Supplementary Fig. 24a), the amount of isolated undesired homopolymer **PDGu_p** after the reaction could be more than 50% of the yield (based on **DGu_p**). This sequential copolymerization of higher reactivity **DGu_p** followed by lower reactivity **BLK_p** could be finished rapidly in <1 h, but achieved only low purity block copolymer with substantial **PDGu_p** homopolymer. We expected that by reversing the order of addition of monomers, i.e. first **BLK_p** and then **DGu_p** (Supplementary Fig. 24b), in which the first block (**PBLK_p**) has a lower reactivity and also a higher transfer rate to **DGu_p**, the block copolymerization would successfully occur. However, the first step requires up to 8 h to reach ~90% conversion (based on **BLK_p**). In addition, the final mixture was very viscous and contained a large proportion of pre-mature terminated **PBLK_p**. Regardless of the sequence of monomer addition, sequential copolymerization cannot successfully synthesize cationic glycosylated block copoly(beta-peptide) with good yield and purity. Unexpectedly, with one-shot AROP with simultaneous feed of the

**Table 1 Design and actual ratios of DGu to BLK before and after deprotection**

| Sample | Design ratio of DGu$_p$ to BLK$_p$ | Actual ratio[a] of DGu$_p$ to BLK$_p$ | Actual ratio[b] of DGu to BLK after deprotection |
|---|---|---|---|
| P1 | **PDGu$_p$(6.7)-*b*-PBLK$_p$(13.3)** | **PDGu$_p$(6)-*b*-PBLK$_p$(14)** | **PDGu(5)-*b*-PBLK(15)** |
| P2 | **PDGu$_p$(8)-*b*-PBLK$_p$(12)** | **PDGu$_p$(8)-*b*-PBLK$_p$(12)** | **PDGu(6)-*b*-PBLK(14)** |
| P3 | **PDGu$_p$(10)-*b*-PBLK$_p$(10)** | **PDGu$_p$(10)-*b*-PBLK$_p$(10)** | **PDGu(7)-*b*-PBLK(13)** |
| P4 | **PDGu$_p$(12)-*b*-PBLK$_p$(8)** | **PDGu$_p$(12)-*b*-PBLK$_p$(8)** | **PDGu(9)-*b*-PBLK(11)** |
| P5 | **PDGu$_p$(13.3)-*b*-PBLK$_p$(6.7)** | **PDGu$_p$(14)-*b*-PBLK$_p$(6)** | **PDGu(10)-*b*-PBLK(10)** |
| P6 | **PDGu$_p$(20)** | **PDGu$_p$(20)** | **PDGu(20)** |

[a]Ratios were calculated based on ${}^{1}$H NMR integrations of (protected) **PDGu$_p$(x)-*b*-PBLK$_p$(y)**
[b]Ratios were calculated based on ${}^{1}$H NMR integrations of (deprotected) **PDGu(x)-*b*-PBLK(y)**

**Table 2 Antimicrobial and hemolytic activity and biocompatibility of (co)polymers**

| Sample | MIC$_{90}$ (µg mL$^{-1}$) | | | | | HC$_{10}$ (µg mL$^{-1}$) RBC[a] | IC$_{50}$ (µg mL$^{-1}$) 3T3[b] |
|---|---|---|---|---|---|---|---|
| | SA 25923 | SA 29213 | MRSA BAA40 | MRSA USA300 | *Bacillus subtilis* 6633 | | |
| **PBLK(20)** | 8 | 8 | 8 | 8 | 4 | 5000 | 18 |
| **PDGu(5)-*b*-PBLK(15)** | 8 | 8 | 8 | 8 | 4 | 3300 | 100 |
| **PDGu(6)-*b*-PBLK(14)** | 16 | 8 | 8 | 8 | 4 | 4800 | 150 |
| **PDGu(7)-*b*-PBLK(13)** | 16 | 8 | 8 | 8 | 4 | >20,000 | 430 |
| **PDGu(9)-*b*-PBLK(11)** | 32 | 16 | 16 | 16 | 8 | >20,000 | 395 |
| **PDGu(10)-*b*-PBLK(10)** | 64 | 32 | 32 | 32/64 | 16 | >20,000 | 630 |
| **PDGu(20)** | >512 | >512 | >512 | >512 | >512 | >20,000 | >1024 |

[a]RBC: red blood cells
[b]3T3: mouse fibroblast cells

**Table 3 Antimicrobial activity against multi-drug-resistant clinically isolated MRSA**

| Serial no. | | Designation | MIC (µg mL$^{-1}$) | | | | Multi-drug resistance | Major lineage/clonal complex[48] |
|---|---|---|---|---|---|---|---|---|
| | | | PDGu(7)-*b*-PBLK (13) | | Resistant antibiotic | | | |
| VAN-resistant *S. aureus* | 1 | HIP11714 | 16 | VAN | 512 | | CIP, CLI, ERY, GEN, LVX, MXF, OXA,RIF, TEC | 5 |
| | 2 | HIP11983 | 16 | | 16 | | CIP, CLI, ERY, GEN, LVX, MXF, OXA, TET | 5 |
| | 3 | HIP13170 | 16 | | 128 | | CIP, CLI, ERY, GEN, LVX, MXF, OXA, TEC, TET | 5 |
| | 4 | HIP13419 | 16 | | 64 | | CIP, CLI, ERY, GEN, LVX, MXF, OXA, TEC, TET | 5 |
| | 5 | HIP14300 | 16 | | 32 | | CIP, CLI, ERY, LVX, MXF, OXA,TEC | 5 |
| | 6 | HIP15178 | 16 | | 512 | | CIP, CLI, ERY, LVX, MXF, OXA, TEC | 5 |
| | 7 | AIS2006032 | 16 | | >512 | | CIP, CLI, ERY, LVX, MXF, OXA, TEC | 5 |
| DAP non-susceptible | 8 | HIP09433 | 16 | DAP | 4 | | CIP, ERY, GEN, LVX, MXF, OXA, PEN, TMP | 45 |
| VAN | 9 | SAMER-S6 | 16 | | 16 | | TMP, PEN, TEC | 5 |
| intermediate *S. aureus* | 10 | 6820 | 16 | | 8 | | OXA, RIF, TEI | 5 |
| | 11 | TTSH-478700 | 8 | | 16 | | CIP, LVX | 22 |
| | 12 | TTSH-671549 | 16 | | 8 | | CIP, ERY, LVX | 22 |
| | 13 | TTSH-478701 | 8 | | 4 | | CIP, ERY, LVX, RIF | 22 |
| | 14 | ATCC 700789 | 16 | | 4 | | CIP, ERY, LVX, RIF, TOB | 5 |
| MDR MRSA | 15 | ATCC BAA38 | 16 | TET | 128 | | PEN, STR | 8 |
| | 16 | ATCC BAA39 | 16 | | 128 | | CIP, ERY, GEN, IPM, LVX, PEN, TMP, TOB | 8 |
| | 17 | ATCC BAA44 | 16 | | 32 | | CIP, ERY, GEN, LVX, PEN, TOB | 8 |

CIP ciprofloxacin, CLI clindamycin, DAP daptomycin, ERY erythromycin, GEN gentamicin, IPM imipenem, LVX levofloxacin, MXF moxifloxacin, OXA oxacillin, PEN penicillin, RIF rifampicin, STR streptomycin, TEC teicoplanin, TET tetracycline, TMP trimethoprim, TOB tobramycin, VAN vancomycin

two beta-lactams, we could achieve successful synthesis of the block copolymers **PDGu$_p$-*b*-PBLK$_p$**.

**PDGu(7)-*b*-PBLK(13) is antibacterial and non-cytotoxic**. The **PDGu(x)-*b*-PBLK(y)** series was tested against a panel of Gram-positive bacteria. The homopolymer **PBLK(20)** was active against most tested bacteria, but was unselective and cytotoxic to eukaryotic cells and also hemolytic (Table 2, Supplementary Figs. 25 and 26). The block copolymerization process decreased cytotoxicity while maintaining potency against *S. aureus*. The copolymer **PDGu(7)-*b*-PBLK(13)** shows the most balanced profile, combining potency against *S. aureus* with good selectivity index (>25) and no hemolysis (HC$_{10}$ > 20,000 µg mL$^{-1}$) (Table 2, Supplementary Figs. 25 and 26). The copolymer shows good activity against USA300 (Table 2), the predominant CA-MRSA[11].

Further profiling demonstrated that the copolymer is also potent against other MRSA strains from major lineages of global epidemiology[48], including (HA-)MRSA strains resistant to multiple conventional antibiotics (including vancomycin, daptomycin) (Table 3). Kill-kinetics experiments revealed that **PDGu(7)-*b*-PBLK(13)** killed replicating MRSA faster than vancomycin (Supplementary Fig. 27). The selection of escape mutants to **PDGu(7)-*b*-PBLK(13)** at 10× its minimum inhibitory concentration (MIC) was unsuccessful, showing that the propensity for emergence of resistance is extremely low (frequency below $3 \times 10^{-10}$, which is much lower than reported values for antibiotics[49,50]). We then tried to select mutants by the continued pressure of sub-inhibitory concentrations of the block copolymer for up to 14 days (as described previously[51]). This approach also did not select for copolymer-resistant MRSA colonies. As a

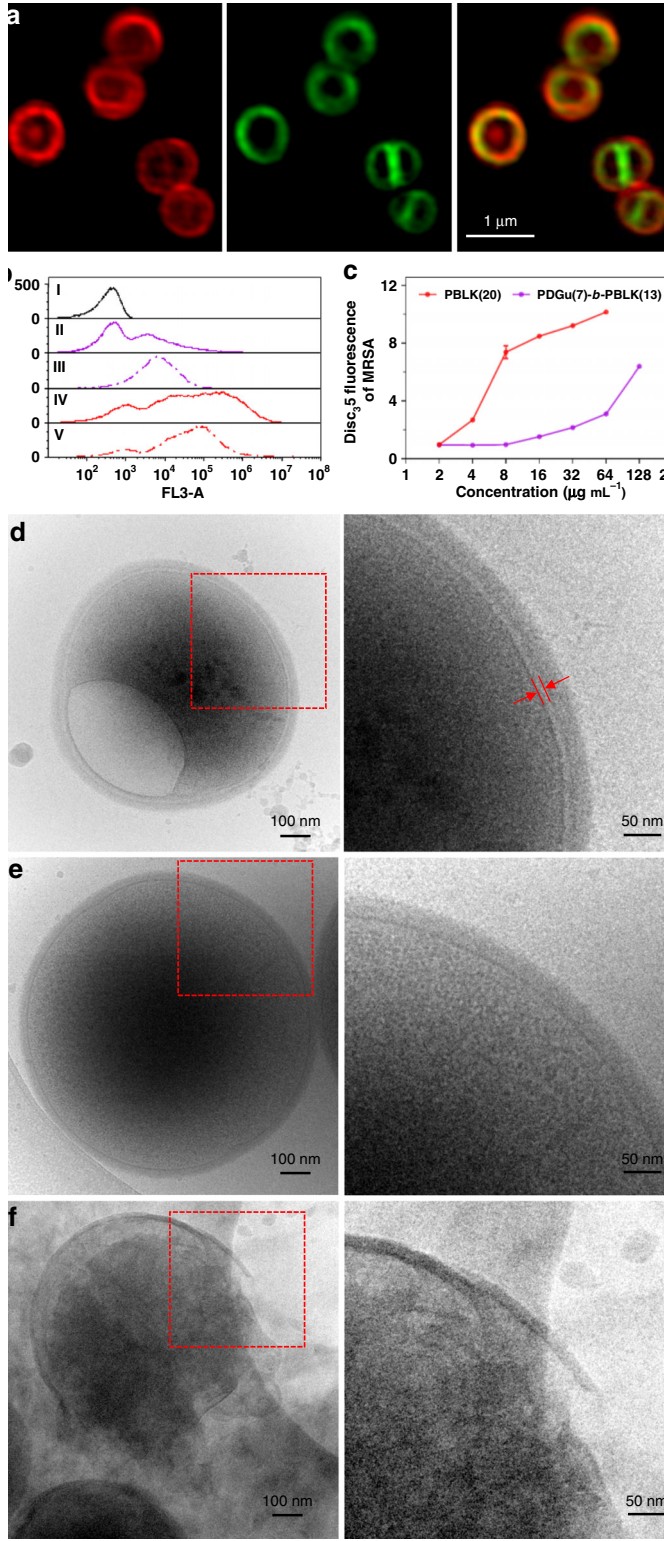

**Fig. 2 PDGu(7)-*b*-PBLK(13) targets bacterial cell envelope. It accumulates in MRSA USA300 cell envelope, mildly (at MIC) permeabilizing the membrane but significantly weakening cell wall/membrane attachment. a** Confocal microscopy images of copolymer-treated MRSA USA300. From left to right: rhodamine-labeled copolymer channel, FM1-43-labeled bacteria membrane channel, superimposed images from both channels, respectively. **b** Flow cytometry study of propidium iodide-stained MRSA USA300. From top to bottom: live bacteria control, bacteria treated with 1× MIC **PDGu(7)-*b*-PBLK(13)**, 4× MIC **PDGu(7)-*b*-PBLK(13)**, 1× MIC **PBLK (20)**, and 4× MIC **PBLK(20)**. **c** DiSC$_3$5 membrane depolarization assay. Data are presented as mean ± standard deviation. **d–f** Cryo-TEM image of polymer treated MRSA USA300. **d PDGu(7)-*b*-PBLK(13)** treated bacteria with enlarged periplasmic space and vacuole structure formation; **e** untreated control; **f PBLK(20)**-treated bacteria with cell lysis

control, escape mutants resistant to the antibiotic ciprofloxacin were easily selected (Supplementary Fig. 28).

**PDGu(7)-*b*-PBLK(13) targets the bacterial envelope.** The structure of the cationic block co-beta-peptide suggests a possible mechanism of action involving membrane interaction. Confocal microscopy of fluorescently labeled bacterial cells showed that the rhodamine-labeled **PDGu(7)-*b*-PBLK(13)**

accumulated preferentially in the bacteria envelope (i.e. cell wall and cell membrane) (Fig. 2a, Supplementary Fig. 29). Membrane damage was confirmed using propidium iodide (PI) as a marker of plasma membrane integrity (Fig. 2b). Results showed that both the block copolymer and cationic homopolymer are membrane active, but the copolymer induces less PI staining, suggesting that it is less membrane-lytic (Fig. 2b). DiSC$_3$5 dye assay, which probes plasma membrane potential changes, corroborated the finding that **PDGu(7)-*b*-PBLK(13)** mildly depolarized the bacterial plasma membrane, unlike the homocationic **PBLK(20)** that had a more pronounced effect (Fig. 2c). Together, the PI staining and DiSC$_3$5 assay results indicate that the copolymer disturbs the bacterial membrane without causing severe leakage.

The effect of **PDGu(7)-*b*-PBLK(13)** on the morphology of *S. aureus* was also visualized by cryo-transmission electron microscopy (cryo-TEM), which revealed a much larger periplasmic space gap (of about 7–8 nm, Fig. 2d; indicated by red arrows), together with bleb and vacuole formation (Fig. 2d, Supplementary Fig. 30). In contrast, periplasmic gap widening, blebs, and vacuoles were not observed in untreated bacteria (Fig. 2e, Supplementary Fig. 31). Treatment with **PBLK(20)** led to significant bacterial envelope deformation, cell leakage, and lysis (Fig. 2f, Supplementary Fig. 32). The copolymer with its hydrophilic sugar block aggregates at the membrane interface, leading to the observed larger periplasmic gap.

Circular dichroism (CD) spectropolarimetry showed that in free solution, the block co-beta-peptide likely adopts a helix–coil conformation attributed respectively to the sugar[52] and cationic[53] blocks (Fig. 3a). However, in the presence of model vesicles containing anionic bacterial lipids, CD spectrum shows that the cationic block of the copolymer, like the cationic homopolymer (**PBLK(20)**), undergoes a transition to likely a left-handed helix structure[54] (Fig. 3b, Supplementary Fig. 33a–f). Computer simulation shows that the binding of **PDGu(7)-*b*-PBLK(13)** to bacterial membrane is provided mainly by the **PBLK** block while the **PDGu** block protrudes into the water–membrane interface because of its weaker binding to the membrane and its strong hydrophilicity (Fig. 3c, Supplementary Fig. 34). In free solution, electrostatic repulsion between the lysine side chains of the cationic block causes the cationic block to exist as a random coil conformation (Supplementary Fig. 34a–d). At the anionic bacterial lipid surface, the positive charges in the **PBLK** block of the copolymer are neutralized by anionic bacterial lipids so that the lysine side chain charge–charge repulsion causing the distortion of the helical conformation of the copolymer **PBLK** block is substantially reduced and the copolymer transitions from a helix–coil structure to a helix–helix structure (Fig. 3d, Supplementary Fig. 34e–g). The resulting helix–helix structure

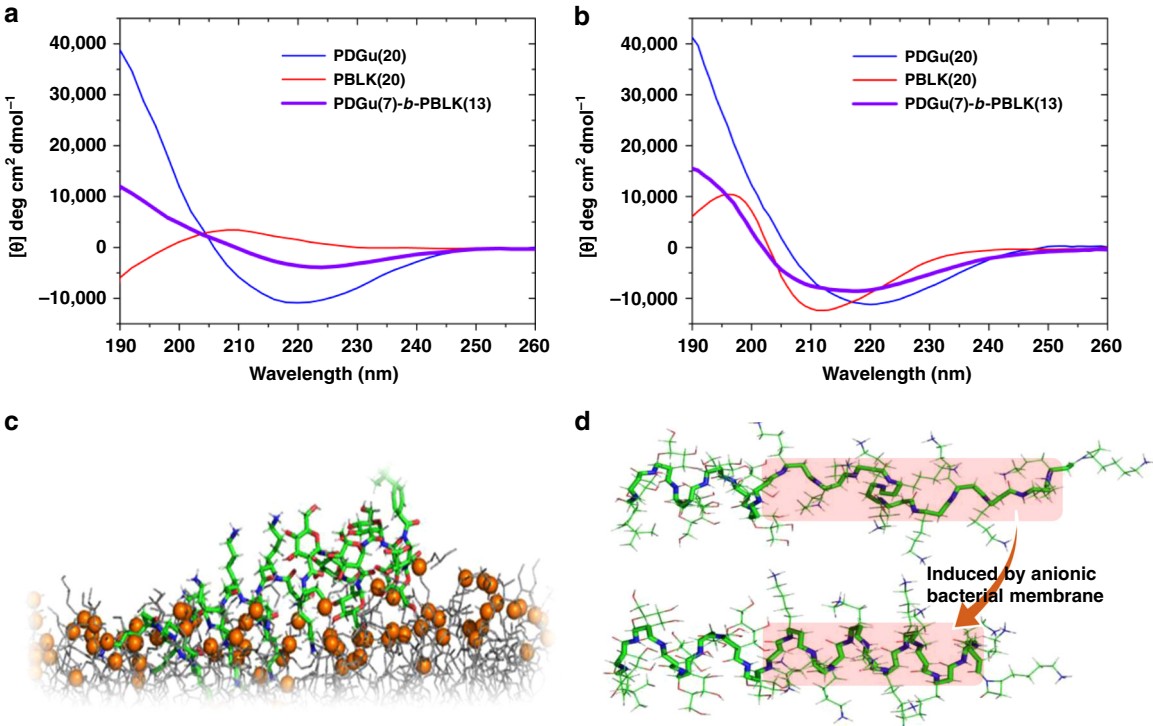

**Fig. 3** Bacterial-induced secondary structure transition of **PDGu(7)-b-PBLK(13)**. **a**, **b** Molar ellipticity [θ] CD spectra of **PDGu(20)** (blue), **PBLK(20)** (red), and **PDGu(7)-b-PBLK(13)** (purple) in DI (**a**) and in the presence of anionic POPG liposomes (**b**). **c** A snapshot of computer simulation of **PDGu(7)-b-PBLK (13)** binding to anionic bacterial membrane. The membrane model is colored as gray lines with the head groups of the lipid molecules shown as orange spheres. **PDGu(7)-b-PBLK(13)** is shown as a stick model, its carbon, oxygen, nitrogen, and hydrogen atoms are colored as green, red, blue, and white, respectively. **d** Computer simulation of transition from helix–coil in solution to helix–helix induced by anionic membrane

of the sugar-cationic block copolymer binds to the anionic bacterial membrane (Supplementary Fig. 34h–k). It is known that beta peptides containing cyclic beta-amino acids adopt different helical structures to those containing non-cyclic amino acids[39]. From the CD data, we see that the (cyclic) **PDGu** spectrum exhibited a minimum at 220 nm while the (non-cyclic) **PBLK** spectrum exhibited a minimum at 213 nm (Fig. 3b). Our computer simulation corroborated that the **PDGu** and **PBLK** blocks adopt different helical conformations, with 3.5 residues/turn and 3 residues/turn, respectively (Supplementary Fig. 34j, Supplementary Note). (In the presence of zwitterionic lipids (model vesicles representative of mammalian membrane), the copolymer retained its helix–coil conformation (Supplementary Fig. 33c)).

The accumulation of the block co-beta-peptide at the outer leaflet of the cytoplasmic membrane causes the increased periplasmic space visible in the cryo-TEM (Fig. 2d), which led to detachment of the cell wall from the cytoplasmic membrane and a weakened membrane–cell wall interface. The copolymer also aggregates inside the cell wall leading to defects in the cell wall function (Fig. 2a, d). The blebs observed with copolymer treatment using cryo-TEM (Supplementary Fig. 30) may be formed by membrane-bound cytoplasm herniating through cell wall defects as intracellular water expands during the freezing process of the cryo-TEM preparation[55,56]. The vacuoles observed (Supplementary Fig. 30) may be ice pockets formed during the cryo-TEM process as water migrates to the polymer-rich periplasmic space since the cytoplasmic membrane is detached from the cell wall. Taken together, the copolymer disturbs the cell envelope which includes the membrane, the membrane–cell wall interface, and also the cell wall.

PDGu(7)-b-PBLK(13) **eradicates persister bacteria and biofilms**. Classical antibiotics are usually significantly less potent against persisters/non-replicating bacteria[1,2,6,7]. Since **PDGu(7)-b-PBLK(13)** kills *S. aureus* by surface contact-induced membrane/envelope damage, we hypothesized that the block copolymer may retain potency against persisters and *S. aureus* biofilms[18]. Nutrient-starved persisters were generated by passaging *S. aureus* in PBS medium, a condition under which the bacteria can survive for extended periods of time without replicating. Consistent with published literature[1], non-replicating *S. aureus* was phenotypically resistant to antibiotics from various categories (including vancomycin, oxacillin, rifampicin, etc.) up to a dose 100× MIC (Fig. 4a). Conversely, **PDGu(7)-b-PBLK(13)** was highly potent against non-replicating starved persister *S. aureus* at a concentration as low as twofold its MIC (Fig. 4b). Furthermore, **PDGu(7)-b-PBLK(13)** effectively eradicated antibiotic-induced persisters that escaped killing by 10× MIC gentamicin and ciprofloxacin treatment (Fig. 4c, d). **PDGu(7)-b-PBLK(13)** was also effective at dispersing preformed MRSA biofilms, achieving a reduction of more than 99.9% of the biofilm bacteria, greatly outperforming vancomycin, which had an insignificant effect on biofilm bacteria (Fig. 4e).

In addition to killing bacteria in biofilm, the block copolymer effectively dispersed the biofilm itself (as shown by confocal microscopy and FESEM) and the dispersed bacteria were shown to be dead (Fig. 4f, g, Supplementary Fig. 35a, b). The homocationic **PBLK(20)** kills biofilm bacteria but does not disperse them (Fig. 4e, h, Supplementary Fig. 35c). Copolymer aggregation at the cell wall accounts for its ability to detach bacteria from biofilm biomass since the sugar block would form a non-fouling coating around the bacteria. (CA-)MRSA USA300

maintained in a broth medium supplemented with glucose typically forms biofilm involving cell-wall anchored protein (fibronectin-binding proteins)[57], whilst many (HA-)MRSA[58,59] and *Staphylococcus epidermidis*[60,61] strains form biofilms involving the polysaccharide intercellular adhesin encoded by the *ica* locus[62]. To determine if the block copolymer is active against other types of biofilms, biofilms formed by various HA-MRSA and methicillin-resistant S. *epidermidis* (MRSE) strains under conditions promoting the *ica* locus expression[63] were treated with the copolymer. Our copolymer **PDGu(7)-*b*-PBLK(13)** was more active than vancomycin in eradicating the biofilms of HA-MRSA and MRSE strains (Fig. 5). Hence, our copolymer is effective not only against MRSA biofilms involving fibronectin-binding protein, but also against other major types of biofilm formed by HA-MRSA and MRSE.

**Copolymer is efficacious in murine and ex vivo human skin models**. Before in vivo efficacy testing, acute toxicity of the block copolymer was evaluated in mice. Intravenous injection of **PDGu (7)-*b*-PBLK(13)** at a cumulative dose of 70 mg kg$^{-1}$ (10 mg kg$^{-1}$ per day × 7 days) was well tolerated in all mice, with no death observed up to 7 days post-injection (Fig. 6a). **PDGu(7)-*b*-PBLK (13)** induced no liver and kidney toxicity, confirming its low in vivo acute toxicity (Fig. 6b, Supplementary Fig. 36, Supplementary Table 3).

The in vivo efficacy of **PDGu(7)-*b*-PBLK(13)** was then evaluated in a mouse model of acute systemic infection. Mice were infected with MRSA USA300 at a lethal dose (100% death within 24 h in untreated controls). At 2 h post infection, a single 5 mg kg$^{-1}$ dose of intraperitoneally (i.p.) injected copolymer resulted in 100% rescue of the mice (6/6 mice) and significantly reduced bacterial loads in major organs (Fig. 6c, d, Supplementary Fig. 37). In contrast, vancomycin treatment at the same dosage achieved only 67% survival (4/6 mice). We further evaluated the efficacy of the copolymer against persisters/biofilm with a deep-seated thigh infection model in neutropenic mice known to be particularly resistant to antibiotic treatment[18,64]. In this model, the copolymer achieved a 93.7% (1.2 log$_{10}$) reduction in bacteria load, whereas vancomycin was ineffective (Fig. 6e). We also evaluated the efficacy of the co-beta-peptide against biofilm bacteria in a murine excision wound model. A biofilm was established in the wound with a 72-h infection period, by which time the bacteria have developed stable biofilms[65,66]. After the 72-h infection, copolymer treatment was given and achieved 99.87% (2.9 log$_{10}$) reduction in bacterial load, which was significantly better than vancomycin (83.8%, i.e. 0.8 log$_{10}$, reduction) (Fig. 6f), showing that the block copolymer has high activity even against an established S. *aureus* infection known to be recalcitrant to antibiotic treatment.

In addition to the murine models, we also demonstrated the efficacy of the copolymer with an ex vivo human skin model with severely established (48 h) infection (Fig. 6g). The copolymer treatment achieved 99.998% (4.6 log$_{10}$) reduction of bacterial burden in contrast to the 97.3% (1.6 log$_{10}$) reduction of vancomycin treatment. Copolymer treated ex vivo human wound sites were also clear of pus/debris corroborating its anti-fouling/biofilm dispersal properties (Supplementary Fig. 38).

## Discussion

Eradication of persisters and biofilms remains one of the biggest challenges in antibacterial drug discovery. Antibiotic-tolerant bacteria are associated with longer treatment time and relapse of infection. Because most antibiotics target macromolecular machinery only essential for active replication, they are significantly less potent against non-replicating persisters or established biofilms. **PDGu(7)-*b*-PBLK(13)** kills non-replicating, antibiotic-tolerant persisters, and biofilm-associated MRSA, both in vitro and in vivo. We show that it can eradicate the clinically relevant CA-MRSA (USA 300). We also showed that our copolymer is just as effective against HA-MRSA strains with resistance to multiple conventional antibiotics (Table 3, Fig. 5). Multi-drug-resistant (MDR) HA-MRSA bacteria cause the majority of nosocomial bacteremia/septicemia and device-related infections involving biofilm formation. The ability of our co-beta-peptide to kill all the sub-populations (planktonic, persister and biofilm states) of MRSA bacteria is attributable to its mechanism(s) of kill —membrane disruption and interface weakening effects which are not related to metabolism. The reduced tendency of the block copolymer to bind mammalian membranes is linked to their less negatively charged surface[36,67]. This co-beta-peptide shows eradication of persister and biofilm MRSA and has ultra-low toxicity, both of which were shown using in vivo murine models. Further, it also shows eradication of an established infection with an ex vivo human skin model.

Upon surface-contact with bacterial membrane, the cationic block undergoes transition from a random coil in free solution to a helix. The block copolymer possesses a unique bacteria-triggered surfactant effect that contributes to biofilm dispersal— the cationic block adsorbs onto the negatively charged bacterial envelope while the hydrophilic sugar block has a strong tendency to promote dissolution, resulting in a "surfactant-like" solvation of bacteria from biofilm. The block copolymer forms an anti-fouling **PDGu** layer around the bacteria. Conversely, the homo-cationic **PBLK(20)** led to pore formation (Fig. 2b, c, f), like other AMPs[36], but without promoting biofilm detachment (Fig. 4h); this is probably linked to the inability of the homocationic polymer to form an anti-fouling layer around the bacteria. The amine group of the cationic block dominates the interactive topology with erythrocytes but its hydrophilicity minimizes hemolysis. Further, the neutral sugar block also increases the hydrophilicity of the block copolymer. Other antimicrobial peptides and biosurfactants (such as surfactin, rhamnolipid, or phenol-soluble modulins) are intrinsically amphiphilic with exposed hydrophobic domains in free solution and are typically hemolytic since their freely exposed hydrophobic moieties would interact with erythrocytes[68].

Biofilm eradication using conventional antibiotics is typically challenging[69,70]. The limited efficacy of vancomycin against biofilms is not an exception; many other antibiotics that are commonly used for MRSA infection have significantly reduced efficacy against biofilm bacteria[71]. Besides antibiotics, our copolymer outperforms many cationic antimicrobial peptides and conventional antiseptic therapeutics in established wound infections that have been previously reported[19,72], not to mention its superior safety profile that makes it suitable for translation into clinics. The copolymer eradicates biofilm MRSA and also disperses the biomass. Since the block copolymer forms an anti-fouling **PDGu** coating around the bacterial cell envelope, the adhesion of the bacteria to extracellular polymeric substances (EPSs) and to substrates is reduced, explaining the strong dispersing effect of **PDGu(7)-*b*-PBLK(13)** on biofilms (Fig. 4e). The coated bacteria can effectively detach due to reduced surface hydrophobicity and interaction with biofilm matrix, leading to biofilm dispersal[73–75]. Moreover, the copolymer biofilm eradication effect is observed in the major types of biofilms formed by different MRSA (and MRSE) strains under various conditions, which include the types involving fibronectin binding protein as well as polysaccharide intercellular adhesin. This is clinically significant since MRSA is a common pathogen that forms biofilms during infection, as well as on medical devices[15,76].

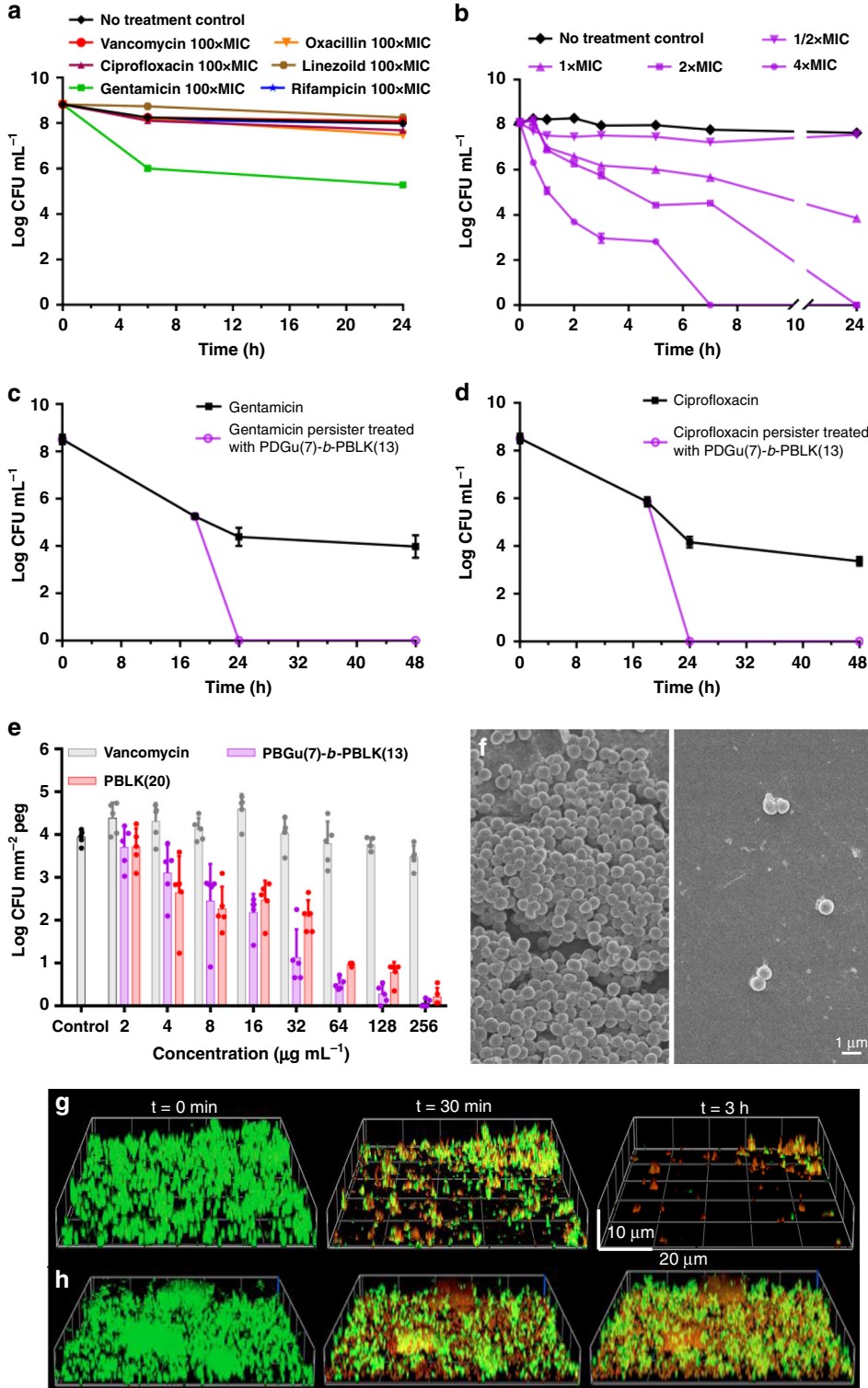

**Fig. 4 PDGu(7)-*b*-PBLK(13) is bactericidal toward MRSA USA300 persisters and biofilms in vitro. a** Kill-kinetics of various antibiotics at 100× MIC; and **b** PDGu(7)-*b*-PBLK(13) against non-replicating MRSA USA300. **c**, **d** Kill-kinetics of **PDGu(7)-*b*-PBLK(13)** at 4× MIC against persisters generated by 10× MIC gentamicin (**c**) and ciprofloxacin (**d**) treatment. **e** Activity of **PDGu(7)-*b*-PBLK(13)** and **PBLK(20)** on established MRSA biofilms using the MBEC™ Assay. Data are presented as mean ± standard deviation. **f** FESEM image of MBEC™ microtiter plate pegs: (left) control peg without treatment and (right) peg treated with **PDGu(7)-*b*-PBLK(13)**. **g**, **h** Confocal microscopy images of **PDGu(7)-*b*-PBLK(13)** (**g**) and **PBLK(20)** (**h**) treated MRSA biofilm at $t = 0$ min, 30 min, and 3 h. Biofilms were stained with Live/Dead BacLight™ kit

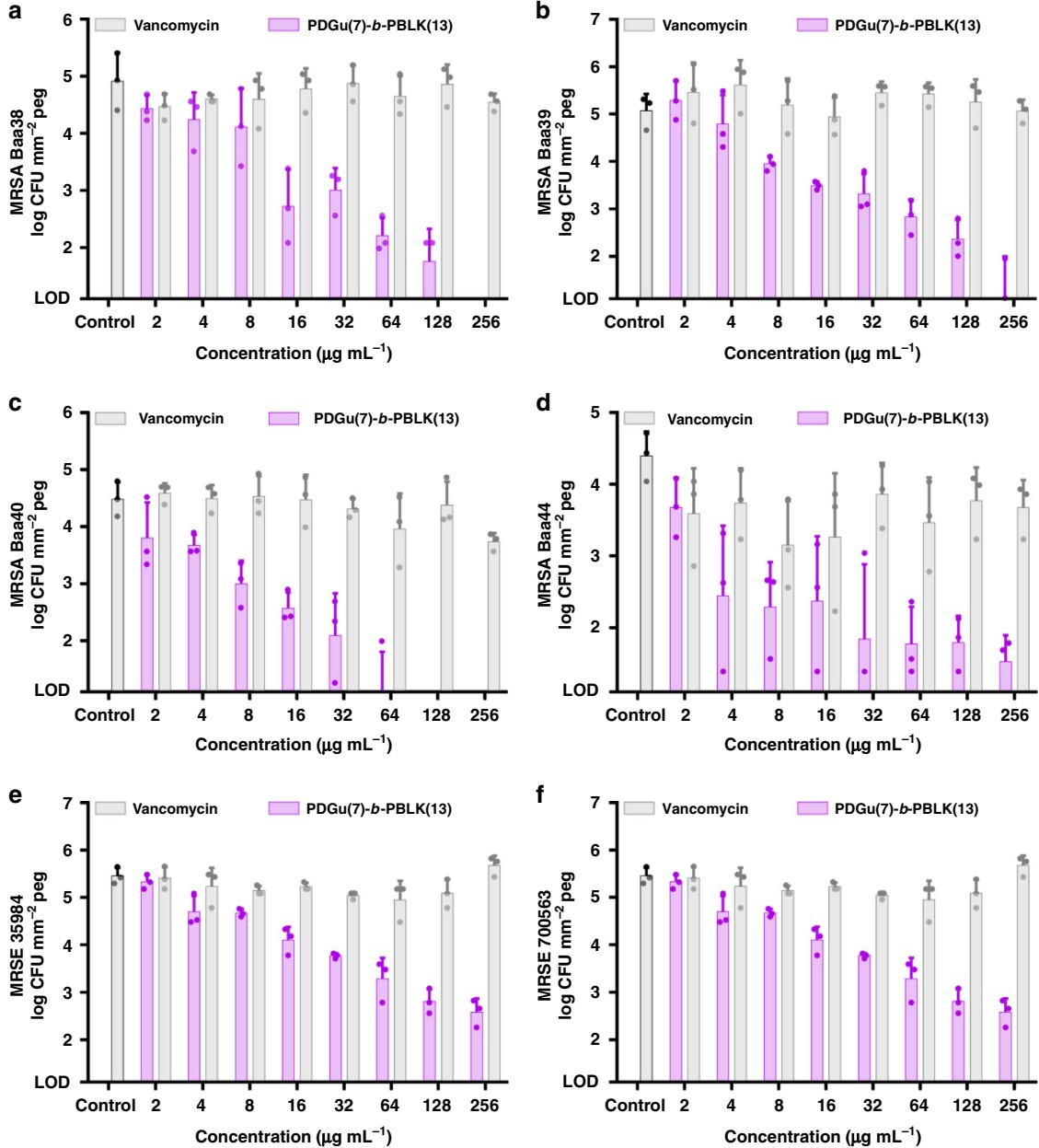

**Fig. 5 PDGu(7)-*b*-PBLK(13)** eradicates biofilms of HA-MRSA and MRSE strains. It shows dose-dependent eradication of biofilm bacteria under conditions that promote polysaccharide intercellular adhesion; *y*-axis: biofilm bacteria (CFU mm$^{-2}$ peg) formed by different HA-MRSA strains (**a** ATCC BAA38, **b** ATCC BAA39, **c** ATCC BAA40, **d** ATCC BAA44) and MRSE strains (**e** ATCC 35984, **f** ATCC 700563). (Vancomycin is used as antibiotic control.) Data are presented as mean ± standard deviation

The block co-beta-peptide (**PDGu(7)-*b*-PBLK(13)**) demonstrates excellent bactericidal efficacy against all MRSA subpopulations, i.e. replicating, biofilm-associated, and antibiotic-induced persister bacteria. It is active against CA-MRSA (USA300) and numerous other MDR HA-MRSA. The cationic block co-beta-peptide undergoes a bacterial-membrane-triggered conformation change from a random coil to likely a helix. Its antibacterial activity in established MRSA murine infection models is superior to that of vancomycin, and it exhibits no acute in vivo toxicity in repeated dosing studies at levels above those required for therapeutic efficacy. Further, the copolymer effectively eradicates established MRSA infections in an ex vivo human skin model. It also kills biofilm bacteria while effectively dispersing the biofilm mass of CA-MRSA; it also shows efficacy against the major types of biofilms formed by HA-MRSA. It acts

as a bacteria-triggered surfactant leading to biofilm dispersal. As resistance toward all classes of antibiotics rapidly evolves and spreads[77], the outstanding efficacy of **PDGu(7)-*b*-PBLK(13)** against *S. aureus* persisters and biofilms, as well as its excellent safety window, makes this block co-beta-peptide a valuable candidate to treat MRSA infections.

## Methods

**General procedure for the polymerization of β-lactams.** In a nitrogen-purged glovebox, a mixture of two β-lactams (**BLK$_p$** and **DGu$_p$**) dissolved in tetrahydrofuran with a defined molar ratio was placed into a Schlenk tube equipped with a magnetic stirrer (Fig. 1a). Then, 4-*t*-butylbenzoyl chloride (tBuBzCl, 5 mol% with respect to the total amount of β-lactam) was added. The Schlenk tube was sealed, removed from the glove box, and cooled to −30 °C under argon atmosphere. To the stirring reaction solution was then slowly added a premade stock solution of lithium bis(trimethylsilyl)amide (LiHMDS, 12.5 mol% with respect to

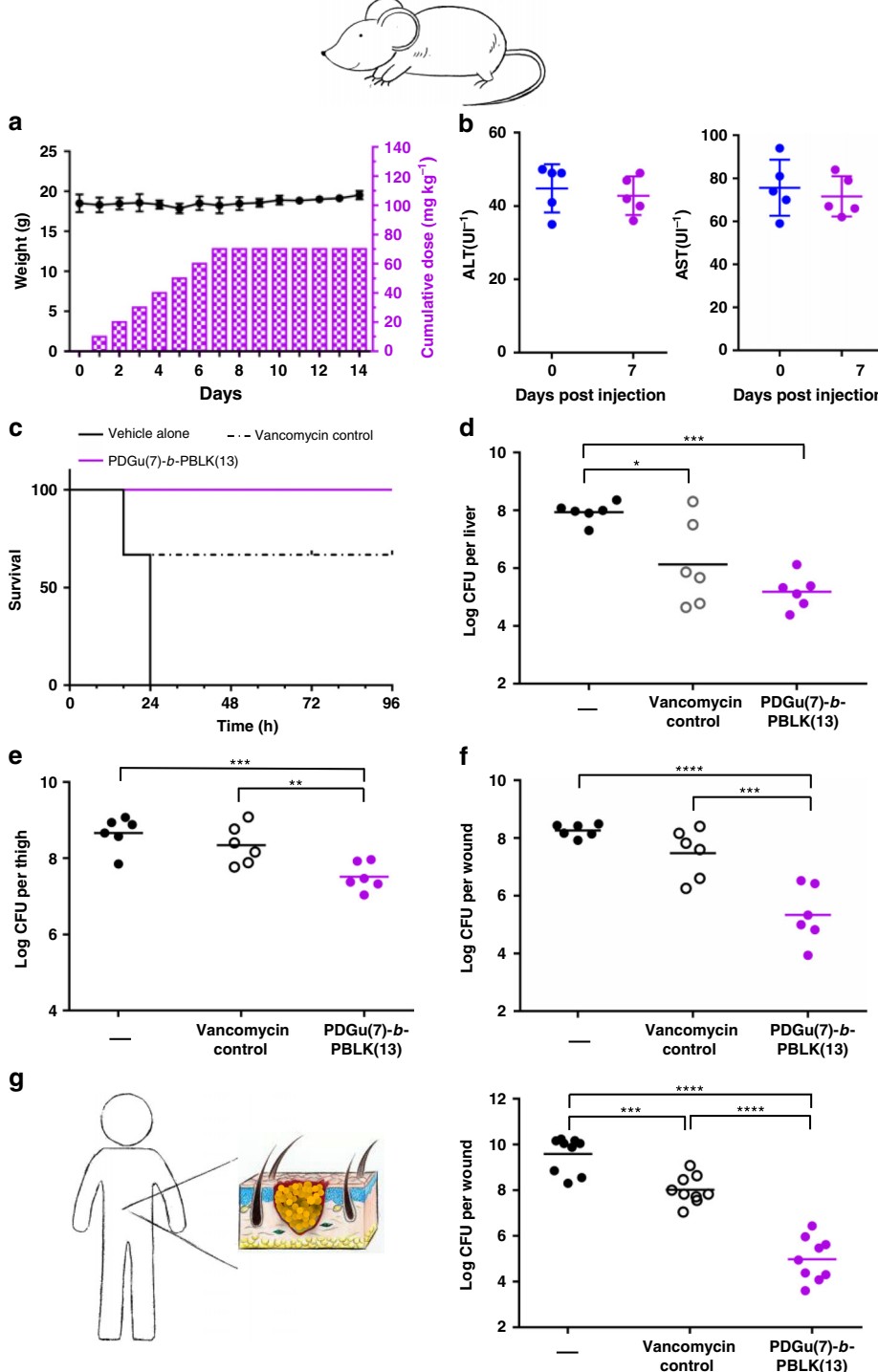

**Fig. 6 PDGu(7)-*b*-PBLK(13) is efficacious in vivo against MRSA USA300 with no toxicity. a, b** In vivo repetitive toxicity of daily 10 mg kg$^{-1}$ i.v. injection of **PDGu(7)-*b*-PBLK(13)** for 7 consecutive days. **a** Mice weight (left *y*-axis) and cumulative dosage (right *y*-axis) over 14 days. **b** ALT and AST biomarker changes at *t* = 0 and 7 days. Data are presented as mean ± standard deviation. **c** Survival% and **d** bacteria log reduction in liver in a systemic infection model. Vehicle alone (–), **PDGu(7)-*b*-PBLK(13)**, or vancomycin control at 5 mg kg$^{-1}$ were applied at a single dose, 2-h post infection. **e** In vivo antimicrobial activity of **PDGu(7)-*b*-PBLK(13)** against MRSA USA300 in a deep-seated neutropenic thigh infection model. First treatment was applied 24-h post infection at 20 mg kg$^{-1}$, with a second dose at 20 mg kg$^{-1}$ applied 3 h later. **f** In vivo antimicrobial activity of **PDGu(7)-*b*-PBLK(13)** against MRSA USA300 in an established murine excision wound model. Vehicle alone (–), **PDGu(7)-*b*-PBLK(13)**, or vancomycin control at the same dosing (i.e. 2.5 mg kg$^{-1}$) were applied six times over 2 days, starting 72-h post infection. **g** Ex vivo antimicrobial activity of **PDGu(7)-*b*-PBLK(13)** against MRSA USA300 in an established wounded human skin model. Vehicle alone (–), **PDGu(7)-*b*-PBLK(13)**, or vancomycin control at 100 µg were applied three times with 3-h interval between treatments, starting 48 h post infection; **p ≤ 0.01, ***p ≤ 0.001, ****p ≤ 0.0001 by one-way ANOVA followed by Dunnett test

the total amount of $\beta$-lactam). The resulting mixture was stirred at $-30\,^\circ\text{C}$ for about 8 h until the reaction was finished (monitored by TLC) and was then quenched with methanol. After completion, a white solid was precipitated by adding hexane (40 mL). The mixture was centrifuged and the supernatant solution was decanted. After two more repetitions of the precipitation/centrifugation procedure, the white pellet was dried overnight under a nitrogen stream to yield the protected product **PDGu$_p$(x)-b-PBLK$_p$(y)** as a white powder.

**General procedure for the debenzylation of PDGu$_p$(10)-b-PBLK$_p$(10)**. Polymer **PDGu$_p$(10)-b-PBLK$_p$(10)** (145 mg) and 54 mg (0.48 mmol, ~1.2 equiv. to monomers) of potassium *tert*-butoxide (KOt-Bu) were dissolved in 5.0 mL of tetrahydrofuran. The polymer solution was added dropwise to a rapidly stirred solution of sodium (160 mg, 7.0 mmol) in liquid ammonia (15 mL) at $-78\,^\circ\text{C}$ under nitrogen. The reaction mixture was warmed to $-55\,^\circ\text{C}$ and maintained at this temperature for about 2 h, after which a saturated aqueous solution of ammonium chloride (NH$_4$Cl, 10 mL) was added to quench the reaction. The solution was warmed to room temperature in a water bath to evaporate the ammonia. The resulting clear solution was filtered, washed with DI water, and dialyzed with 1000 MWCO tubing for 36 h with ten water changes. After lyophilization, **PDGu(7)-b-PBLK(13)** was obtained as an amorphous white solid. Copolymers **(PDGu$_p$(x)-b-PBLK$_p$(y))** with other design block lengths (x, y) were synthesized by a similar procedure.

**Reaction kinetics studies**. High-performance liquid chromatography (HPLC) was employed to determine $\beta$-lactam consumption. For these measurements, a series of reactions was performed with identical conditions (temperature: $-30\,^\circ\text{C}$, initial concentration: [**DGu$_p$**] = 0.05 M, [**BLK$_p$**] = 0.05 M, activator [ArCOCl] = 0.005 M, inititator [LiHMDS] = 0.0125 M) but quenched at different times. After purification by flash column chromatography, concentrated reaction mixtures were mixed with a certain amount of paraben (internal standard) and diluted with acetonitrile to the same volume. Aliquots of these solutions were transferred to vials and injected into a Shimadzu LC-20AD HPLC workstation equipped with an IB column. Monomer concentration was calculated from the peak area ratio relative to a known amount of internal standard. GPC curves were determined versus polystyrene standards using dimethylformamide (1 mg mL$^{-1}$ LiBr) as the eluent at a flow rate of 1.0 mL min$^{-1}$ through two Styragel columns (HR5 and HR5E, 7.8 × 300 mm) in series at $40\,^\circ\text{C}$ with a refractive index detector.

**Bacterial strains**. All bacteria strains of Table 2, Strains #14–17 of Table 3, and all bacteria strains of Fig. 5 were purchased from ATCC. Vancomycin-resistant *S. aureus* (Strains #1–7 of Table 3) were kindly provided by Prof. Barry N. Kreiswirth and Dr. José R. Mediavilla from the Center for Discovery and Innovation, Hackensack Meridian Health (USA). Daptomycin non-susceptible vancomycin-intermediate MRSA (Strains #8 and 9 of Table 3) were kindly provided by BEIresources. org. Strain #10 of Table 3 was kindly provided by Dr. Adriana Rosato from the Houston Methodist Research Institute (USA). Strains #11–13 of Table 3 were kindly provided by Tan Tock Seng Hospital (TTSH, Singapore).

Multilocus sequence typing (MLST) characterization for the three VISA strains from local hospital (Strains #11–13 of Table 3) were conducted. Overnight culture from single colony was washed with 10 mM Tris buffer, resuspended in 800 μL of lysis buffer containing 5 mg mL$^{-1}$ lysozyme, 10 mM EDTA, and 10 mM Tris. After 1 h incubation at $37\,^\circ\text{C}$ with shaking, the suspension was heated to $95\,^\circ\text{C}$ for 10 min and subsequently transferred to ice. 1 mL of ice-cold phenol/chloroform/isoamyl alcohol (25:24:1) was added and mixed thoroughly by inverting the tubes five times, followed by incubation for 5 min on ice. After centrifugation at 20,000×g for 20 min, the aqueous layer was transferred to a fresh tube and DNA was precipitated by adding 1 mL of ice-cold ethanol, followed by incubation for 15 min on ice. The DNA pellet was collected by centrifugation and washed once with ice-cold 70% ethanol, and resuspended in 50 μL of water. The extracted DNA was amplified by PCR using Novagen KOD Hot Start DNA Polymerase, and the amplified products were sequenced by Sanger sequencing. The obtained sequence was submitted to MLST database (http://www.mlst.net/) to obtain the sequence type (ST).

**MIC determination**. Bacteria in logarithmic phase of growth were diluted to $1 \times 10^6$ colony-forming units (CFU) per milliliter in Mueller Hinton Broth (MHB, Difco®). Polymers were dissolved at 10.24 mg mL$^{-1}$ in deionized water and diluted to desired concentration in MHB using twofold serial dilution in a 96-well plate (Nunc™). A total of 50 μL of bacteria in MHB suspension were added to 50 μL of polymer to achieve a final volume of 100 μL per well. The plate was incubated aerobically at $37\,^\circ\text{C}$ for 18 h, and the optical density of each well was measured at a wavelength of 600 nm (TECAN, infinite F200). MIC$_{90}$ is defined as the lowest concentration that exhibited more than 90% inhibition of the bacteria growth. All tests were performed three times independently with two samples in each test. For tests involving daptomycin, 50 μg mL$^{-1}$ CaCl$_2$ is supplemented to the medium.

**MTT cytotoxicity test**. Mouse fibroblasts (3T3 cells) were purchased from ATCC. Cells were seeded at $2 \times 10^4$ cells per well in a volume of 200 μL of Dulbecco's Modified Eagle's medium (DMEM, Gibco™) in a 96-well tissue culture plate, and

incubated at $37\,^\circ\text{C}$ in a humidified incubator with 5% CO$_2$ for 24 h. Polymer stock solution was prepared in PBS (phosphate-buffered saline, Gibco™) at a concentration of 10 mg mL$^{-1}$ and diluted to desired concentrations in DMEM complete medium. Polymer in DMEM solution was added into the cell-seeded 96-well plate and incubated at $37\,^\circ\text{C}$ for 24 h. Subsequently cells were rinsed with PBS and 1 mg mL$^{-1}$ MTT in DMEM was added into each well. The plate was incubated for 4 h, after which the MTT solution was aspirated and 100 μL of dimethyl sulfoxide was added into each well. The plate was shaken at 150 rpm for 10 min and the absorbance of each well was measured at 570 nm using a microplate reader spectrophotometer (BIO-RAD, Benchmark Plus). Cell viability was calculated using the following formula and IC$_{50}$ was interpolated using mean values of triplicate measurements.

$$\% \text{ Cell viability} = \frac{\text{Average abs of treated cells}}{\text{Average abs of controls}} \times 100\%. \quad (1)$$

**Hemolysis assay**. The human blood hemolysis experiment was reviewed and approved by the Institutional Review Board of Nanyang Technological University (IRB-2015-03-040). Human blood samples were obtained from a healthy donor (age 23, male) and informed consent was given in accordance with NTU-IRB ethical principles. Fresh human blood was washed with PBS twice and red blood cells were resuspended to 5% v/v in PBS. Polymers were twofold serial diluted in PBS and 50 μL of polymer solution samples were mixed with red blood cell suspension in a 96-well plate. The plate was incubated for 1 h at $37\,^\circ\text{C}$ under mild shaking. The microplate was centrifuged at 1000 rpm for 10 min; 80-μL aliquots of the supernatant were then transferred to a new 96-well microplate and diluted with another 80 μL of PBS. Hemolytic activity was calculated from absorbance measured at 540 nm using a microplate reader spectrophotometer (Benchmark Plus, BIO-RAD):

$$\text{Hemolysis} \% = \frac{O_p - O_b}{O_t - O_b} \times 100\%, \quad (2)$$

where $O_p$ is the absorbance of polymer, $O_b$ is the absorbance of negative control, and $O_t$ is the absorbance of positive control. HC$_{10}$ values (concentration that causes 10% hemolysis) were interpolated using mean values of triplicate measurements.

**Kill kinetics of non-replicating/antibiotic-generated persisters**. A culture of MRSA USA300 was washed two times with PBS and resuspended in PBS at a final concentration of $10^8$ CFU mL$^{-1}$. The bacteria suspension was incubated in PBS for 1 h to adapt the cells to starvation. Polymer and antibiotic were added to 1 mL of bacteria in PBS suspension in Eppendorf tubes to achieve a desired final polymer/ antibiotic concentration. The Eppendorf tubes were incubated aerobically under shaking at $37\,^\circ\text{C}$. At desired time points, 20 μL of each sample was serial diluted in PBS, and plated on nutrient agar plates for CFU determination. For killing of persister bacteria that escaped standard antibiotic treatment, $10^8$ CFU log-phase bacteria in 1 mL of MHB were challenged with antibiotics (ciprofloxacin or gentamicin) at 10× MIC for 18 h. Half of the bacteria were washed to remove antibiotics and challenged with copolymer at 4× MIC in MHB. The other half continued under challenge with antibiotics as a control. Aliquots of samples at each time point were washed with PBS twice to remove antibiotics/polymers and serial diluted in PBS to determine CFU. Error bars were produced from two independent tests, with duplicate samples for each test.

**Spontaneous mutation frequency**. At day 1, initial inocula of $3.5 \times 10^9$ CFU exponential-phase MRSA USA300 in 10 mL of MHB were placed in 50-mL falcon tubes and challenged with polymer at 10× MIC under shaking at $37\,^\circ\text{C}$. Polymer was changed every 48 h during the incubation. The OD$_{600nm}$ values were recorded daily over 6 days. At days 3 and 6, 100 μL of the sample was serially diluted in PBS and plated on nutrient agar plates for CFU determination.

**Resistance evolution by serial passage**. Exponential-phase MRSA USA300 ($10^6$ CFU) were grown in 1 mL of MHB containing copolymer or antibiotic control ciprofloxacin at a gradient of concentrations: 0.25× MIC, 0.5× MIC, 1× MIC, 2× MIC, and 4× MIC. At 24-h intervals, the cultures were checked for growth and the MIC value for each day was recorded. Cultures from the second highest concentrations that allowed growth (OD$_{600} \geq 1$) were diluted 1:1000 into fresh MHB containing 0.25× MIC, 0.5× MIC, 1× MIC, 2× MIC, and 4× MIC of copolymer/ ciprofloxacin. The serial passaging was repeated daily for 14 days. Three independent biological replicates were conducted for each experiment.

**Microscopic studies**. Log-phase MRSA USA300 bacteria were washed and diluted to $10^8$ CFU mL$^{-1}$ in PBS and incubated with polymer at $37\,^\circ\text{C}$ for 4 h. The bacteria suspension was centrifuged and resuspended in PBS for cryo-TEM imaging. For confocal imaging, $10^8$ CFU mL$^{-1}$ log-phase bacteria were incubated with rhodamine-labeled polymer for 1 h and subsequently stained with membrane dye FM1-43FX before confocal microscopy imaging.

**Bacterial membrane integrity assays**. Log-phase bacteria (MRSA USA300) were washed and diluted to $10^8$ CFU mL$^{-1}$ in PBS and incubated with polymer at $37\,^\circ\text{C}$

for 0.5 or 1.5 h, and stained with PI (L13152 Invitrogen). Samples were washed twice and resuspended in PBS to $10^7$ CFU mL$^{-1}$ and analyzed using flow cytometry (BD Accuri C6 plus). Data are plotted as normalized histogram of fluorescence intensity from FL3 channel. For DiSC$_3$5 membrane depolarization assay, logphase bacteria were washed and resuspended to $10^7$ CFU mL$^{-1}$ in 5 mM HEPES buffer (pH 7.8) containing 20 mM glucose and 0.1 M KCl. DiSC$_3$5 solution was added to bacteria suspension to achieve a final concentration of 100 nM and allowed to quench for 30 min. Polymer solution was added to achieve the desired concentration in a black 96-well plate (Costar). Fluorescence readings were recorded 5 min after polymer addition with a Tecan reader at an excitation wavelength of 622 nm and an emission wavelength of 670 nm.

**Biofilm assays.** A total of 150 μL of MRSA USA300 bacteria in tryptic soy broth (TSB) containing 1% glucose (initial inoculum of $10^6$ CFU per well) was added into each well of an MBEC plate (Innovotech, Canada). After 24 h of incubation at 37 °C under mild shaking, the pegs were washed twice using 200 μL of PBS and transferred to a 96-well plate containing a twofold dilution series of polymer in PBS (200 μL per well). MBEC pegs were exposed to the polymer for 3.5 h, and subsequently washed before sonication-releasing the biofilm bacteria into the recovery plate for CFU counting. For FESEM imaging of pegs, untreated control and copolymer-treated (64 μg mL$^{-1}$ for 3.5 h) pegs were removed aseptically, fixed with 4% paraformaldehyde at 4 °C overnight, and dehydrated using a graded ethanol series. For confocal imaging, 24-h preformed biofilm was established in a collagen-coated glass-bottom Petri dish (MatTek). Biofilm was stained with BacLight™ live/dead kit. Polymer in PBS solution (32 μg mL$^{-1}$) was dropwise added to avoid physical disturbances to biofilm and confocal images were taken immediately after polymer addition and at defined times thereafter. For biofilms formed with polysaccharide intercellular adhesin, various strains of HA-MRSA and MRSE biofilms were established under high-salt conditions using TSB + 4% NaCl.

**Secondary structure study.** Polymers were dissolved at 0.05 mg mL$^{-1}$ in different media, i.e. DI water, 10 mM phosphate buffer (pH 2.6–8.7), 20 mM carbonate buffer (pH 10.8) and in the presence of 1 mg mL$^{-1}$ POPG or POPC liposomes. (For the POPG liposomes, the polymer:lipid (P:L) molar ratios of **PBLK(20)**, **PDGu(20)**, and **PDGu(7)-b-PBLK(13)** are 1:79, 1:83, and 1:89, respectively.) CD spectra were measured from 190 to 260 nm with 0.5-nm step size, with each measurement performed twice. The final data are presented as the mean values after background extraction.

**Mouse model of MRSA USA300 infection and in vivo toxicity.** The animal experiments were reviewed and approved by the Animal Ethics and Welfare Committee (AEWC) of Ningbo University. A single 5-mm-diameter excision wound was created on female C57BL6 mice and inoculated with 2.5 μL of MRSA USA300 suspended in PBS ($5 \times 10^5$ CFU mL$^{-1}$). For treatment initiated at 72 h postinfection, treatments were applied in total six times over 2 days, i.e. three times per day with 4 h between each treatment. Samples were harvested 4 h after the last treatment to determine the CFU. For systemic infection, $10^8$ CFU mL$^{-1}$ MRSA USA300 in PBS (with 5% mucin) were i.p. injected into female balb/c mice and the infections were allowed to develop for 2 h. At 2 h post infection, treatment was injected i.p. at 5 mg kg$^{-1}$ in 200 μL of PBS. Twenty-four hours post infection, the mice were sacrificed and i.p. fluid, liver, kidney, and spleen were harvested and homogenized to determine the CFU. For survival test, mice were monitored up to 96 h post infection/treatment. For deep-seated thigh infection, neutropenic ICR mice were infected by injecting 50 μL of stationary-phase MRSA USA300 in PBS ($10^5$ CFU per thigh) into thigh muscles and infection was developed for 24 h. Mice were treated with 20 mg kg$^{-1}$ copolymer or antibiotic subcutaneously; 3 h later, a second treatment was given by the same route. Thigh tissues were harvested 24 h post first treatment and homogenized to determine the CFU. For in vivo toxicity determination, 10 mg kg$^{-1}$ of the block copolymer in 200 μL of PBS was injected into female balb/c via tail vein daily for 7 days. Clinical biomarkers were recorded before, 24 h after, 3 days after, and 7 days after the first injection. For histological analysis, mice were sacrificed at 48 h post final injection and tissues were harvested for H&E staining and examination.

**Ex vivo wounded human skin infection model.** Human skin samples were purchased from Biopredic International. All samples were obtained from healthy donors undergoing cosmetic surgery and informed consent was given in accordance with French law and ethical principles. Five-millimeter-diameter wounds were created and inoculated with 10 μL of MRSA USA300 ($2 \times 10^9$ CFU mL$^{-1}$). Infections were developed for 48 h and wound sites were gently rinsed with PBS to remove planktonic bacteria; PBS vehicle alone, 100 μg of vancomycin, or copolymer were applied three times with a 3-h interval between each treatment. Three hours post last treatment, the samples were harvested and homogenized for CFU determination.

**Reporting summary.** Further information on research design is available in the Nature Research Reporting Summary linked to this article.

## Data availability

The data that support the findings of this study are available from the corresponding authors on request. The source data underlying Figs. 4e, 5a–f, 6a, b and 6d–g, Supplementary Fig. 37a–d and Supplementary Table 3 are provided in the Source Data file.

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

## Acknowledgements

We thank Dr. Adriana Rosato for generously sharing strain 6820. We thank Prof. Angelika Gründling and Dr. Moon Tay for their scientific discussions on the manuscript. We thank the funding support from a Singapore Ministry of Education Tier 3 grant (MOE2013-T3-1-004) and a Singapore Ministry of Health Industry Alignment Fund (NMRC/MOHIAFCAT2/003/2014).

## Author contributions

K.Z., Y.D., K.P., and M.B.C. conceived the project and wrote the manuscript. Y.D., Z.S., and C.R. synthesized the polymers. K.Z., S.R., and L.R. conducted the in vitro biological tests. K.Z., S.J., and D.K. conducted the in vivo tests. K.Z. and M.E.T. conducted the ex vivo human skin test. Y.L. conducted the computer simulation study. K.P. and M.B.C. supervised and guided the overall research. M.B.C. and G.C.B. guided chemical synthesis. J.R. and Y.Z. supervised the in vivo toxicity tests. Y.M. supervised

the computer simulation study. K.C.T. supervised the physical property study. K.M., P.P.D., and O.N. isolated and provided strains from local hospital TTSH and provided useful suggestions. J.R.M. and B.N.K. isolated relevant vancomycin-resistant MRSA strains and conducted the susceptibility tests. H.D., X.L., Y.R.C., and P.T.H participated in the supervision of the project. All authors discussed the results and commented on the manuscript.

## Competing interests

The authors declare no competing interests.
