## [Peer Review File · Nature Communications]

Reviewers' comments:

Reviewer #1 (Remarks to the Author):

The first part of this paper describes chemical synthesis and chemical characterization of novel block co-beta peptide polymers which were found to have quite potent activity against *Staphylococcus aureus*. The second part of the paper describes investigations into the mechanistic basis of the bactericidal effect and demonstrate that the molecule can kill both actively growing as well as persister cells that occur in bacterial populations, in biofilms it results in dispersal and bactericidal activity and has efficacy in mouse models of infection. The lack of toxicity towards mammalian cells and in uninfected animals and lack of development of resistance to 10x the MIC argues well for potential clinical development as an antimicrobial.

1. Apart from the MIC data in Table 1 it appears that most, if not all, the in vitro and in vivo studies were performed with a single MRSA strain USA300. Rather than test a few ill-characterized MRSA strains it is necessary to test activity against well characterized MRSA strains representing the major lineages. In particular it is necessary to demonstrate activity against hospital-associated MRSA strains with resistance to multiple conventional antibiotics (including glycopeptides vancomycin and daptomycin) because these cause the majority of nosocomial device-related infections involving biofilm formation, as well as the majority of nosocomial bacteraemia/septicaemia

2. In vitro biofilm formation. The methods and figure legend do not specify if these experiments were performed with USA300. If they were then the type of biofilm formed under the growth conditions employed (glucose containing broth) is likely to involve cell wall anchored proteins in the accumulation phase (see McCourt J, Fibronectin-binding proteins are required for biofilm formation by community-associated methicillin-resistant *Staphylococcus aureus* strain LAC. FEMS Microbiol Lett. 2014 Apr;353(2):157-64.....). While some HA-MRSA strains form CWA protein-dependent biofilm in vitro a significant proportion of MRSA and also *S.epidermidis* form biofilm that involves the polysaccharide intercellular adhesin encoded by the *ica* locus. Before the authors can claim that their molecule is effective against *S.aureus* biofilm this must be investigated further

3. Daptomycin has been used to treat persistent bacteraemia as an alternative to vancomycin and indeed when vancomycin insensitive strains are involved. Daptomycin resistance involves changes in membrane lipid composition (gain of function MprF mutants) which result in an increase in lysyl-phosphatidyl glycerol and hence an increased positively charged membrane. Does the co-beta peptide have activity in mutants where membrane lipid composition is altered. ?

4. Expression of b-lactam resistance in MRSA requires the integrity of the membrane lipid rafts enriched with staphyloxanthin and containing the membrane associated protein flotillin. Any disruption to the integrity of lipid rafts (and/or PBP2a) results in bacteria becoming susceptible to beta-lactams. There is considerable interest in resurrecting b-lactams to combat MRSA with molecules that act synergistically. Does the co-polymer have synergistic activity with b-lactams against MRSA?

5. Table 1 describes MIC of a "panel of Gram-positive bacteria" but apart from *S.aureus* only *Bacillus subtilis* is shown. Does the molecule have activity against other Gram-positive pathogens such as *Streptococcus pneumoniae* and other streptococci. There is no mention of activity against Gram-negative bacteria. Can the authors provide information on any activity against ESKAPE pathogens?

6. Development of resistance. A large inoculum of *S.aureus* cells $>10^9$ were exposed to 10x MIC for a prolonged period. What happened if bacteria were exposed to the MIC or below the MIC with surviving populations being exposed to slightly increased concentrations?. This approach would allow small incremental increases which would mimic better the situation in vivo. For example

VISA strains emerge after acquisition of 5 or 6 mutations in different chromosomal genes

Reviewer #2 (Remarks to the Author):

This paper describes the synthesis and antimicrobial activity of a novel block co-polymer containing D-glucose and beta lysine. This block polymer was designed to be amphipathic, containing negatively charged glucose side chains at the N-terminus and positively charged residues at the C-terminal end. In this way, it was hypothesised that the glucose side chains would facilitate penetration into the peptidoglycan wall and the lysine residues would mediate binding and disruption of the inner phospholipid membrane bilayer.

A significant amount of work has been presented. The results demonstrate good antibacterial activity and low host toxicity which is a challenge to achieve and demonstrates the potential of these materials. However, there are a number of issues which I feel diminish the novelty and quality of the studies and which need to be addressed before this manuscript can be considered for publication. In particular, there are significant issues with the CD analysis, the computer simulation and referencing of prior work.

Page 4, Line 69, the authors state "beta-peptides have relatively hydrophobic backbones". While beta peptides have an extra methylene in the backbone and therefore additional hydrophobicity related to these addition atoms, the relative hydrophobicity of beta peptides is more controlled by the structure of the side chains. This statement should be re-stated.

The authors need to take greater care in accurate citation:

Page 3, line 66. References 23-25 are incorrect. Ref 23 is a cyclic alpha-peptide with a beta hairpin structure. Ref 24 is a review on beta-peptoids and the authors should refer to primary papers as specific examples. Ref 25 is a large review on the biochemistry and medicinal chemistry of the dengue virus protease – again the authors should cite the specific paper relevant to their paper.

Page 5. The authors need to refer to much earlier studies which reported the synthesis of nylon-3 and analogues together with structural analysis. These papers are from the group Munoz-Guerra, Subirana et al and include (amongst at least 8) Nature, 1984, 311, 53-54; Macromolecules 1998, 31, 124-134; Macromolecules 1987, 20, 62-68. The authors need to state how their work differs and improves on these earlier studies.

The review by Konai et al [Biomacromolecules (2018) 19, 1888–1917] provides a very detailed overview of the application of polymers with a section on nylon-3/poly-beta lactams. Again, how does the chemistry and design presented by the authors differ and/or improve on what has been previously published?

The key papers of the Gelman group should also be cited: Porter EA, Weisblum B, Gellman SH. J Am Chem Soc. 2002 Jun 26;124(25):7324-30. Porter EA, Wang X, Lee HS, Weisblum B, Gellman SH. Nature. 2000 Apr 6;404(6778):565.

Page 5, line 104. The authors state "The molecular weight of the product increased monotonically..". Do the authors mean linearly?

Page 5. The authors refer to estimated molecular weights based on the GPC retention. Why did they not use mass spectrometry to determine accurate mass?

Page 6. If NMR etc suggests a ratio of G7:K13, why do the authors persist in referring to the

product as 10:10 – this was the target ratio – but using this nomenclature throughout the manuscript gives the wrong impression that this is indeed the precise composition.

Page 6, line 120. “corroborating their blocky rather than random structures”. Should be “block”, not “blocky”. Also on line 140.

The antibacterial activity appears promising – why was the vancomycin dose of 2.5mg/kg used in the in vivo studies ? This needs to be explained as the authors claim that their peptide is superior to vancomycin.

The use of the word “probably” is inappropriate – the authors need to be more circumspect in their conclusions – 3 examples are listed below:

Page 8, line 175. “copolymer with its hydrophilic sugar block probably aggregates at the membrane interface, leading to the observed larger periplasmic gap.”.

Page 9, line 199, “The accumulation of the block co-beta-peptide at the outer leaflet of the cytoplasmic membrane probably causes the increased periplasmic space visible in the cryo-TEM (Fig. 2d),”

Page 9, line 209, “copolymer disturbs the cell envelope which includes probably the membrane, the membrane-..”

There are a number of issues with the CD analysis:

The CD spectra shown in Fig 3c and 3d show the structure in water and POPG respectively of PDGu(20), PBLK(20) and the 10:10 co-block peptide. Fig 3c and d indicates that both PDGu(20) and the 10:10 peptide adopt secondary structure in water and in POPG, but PBLK only adopts structure in POPG liposomes.

In addition, Fig S10 e, f and g show the CD spectra of these 3 peptides at 3 different pHs, water and POPG and POPC. This data further demonstrates the persistence of the helical structure for PDGu(20) in all conditions while PBLK(20) only adopts structure in both lipids. The block copolymer adopts structure in all solutions, indeed, the spectra appear to be composite in shape of the spectra in Fig S10 e and f.

Based on spectra published on other beta-peptides, the authors conclude the presence of a left-handed helix as beta-peptides containing non-cyclic beta amino acids adopt helical structures which are left-handed in conformation. However, beta-peptides containing cyclic beta-amino acids adopt different helical structures to beta peptides containing non-cyclic amino acids. This is quite evident when one compares the spectral minima for PDGu(20) [$\sim 222\text{nm}$] and PBLK(20) [$208\text{-}210\text{nm}$]. The authors need to take great care to not over-interpret the spectra.

In addition, the authors state on page S38 that PBLK(20) interacts strongly with anionic lipids because of the change in spectra from an extended coil in the absence of lipid to a helical structure in POPG. However spectral changes do not provide any information on binding affinity, only partial information on structural changes between different solutions. This statement needs to be changed.

Moreover, spectra were obtained at a peptide concentration of 0.05 mg/ml (and re-stated as 50 $\mu\text{g/ml}$ elsewhere – need to be consistent) - given that the mass of each peptide is different, the changes in spectral intensity cannot be directly related to degree of secondary structure between peptides in any solution as molar ellipticity has not been calculated.

What is the peptide:lipid (P:L) ratio in the CD experiments with liposomes? This is an important parameter to analyse the results especially since one of the main conclusions is that the lipids induce structure. However, the P:L ratio also influences the structure change.

There are a number of issues with the modelling work:

Choice of parameters. The authors state that they use the CHARMM general force field to model the beta peptides. Parameters for beta peptides are available in CHARMM, but if they used those parameters, they should cite Rathore et al. 2006 (doi: 10.1529/biophysj.106.084491). If they indeed just used the general CHARMM force field, they need to provide more details and justification. Particularly regarding the treatment of the backbone.

The composition of the peptides used for simulations – any conclusions based on modelling of either block of 10 or a 10:10 co-block is a rough approximation without accurate mass or structure.

Model membrane – which phospholipids were used for creating the model bilayer? There are no details apart from a comment about a negative lipid. Again, the results are not valid. Distance between beta peptide and membrane. What exactly is the distance in Figure S10a? Center of mass? How far from the membrane was the peptide at the start of the simulation? And in what orientation?

Sampling. Three repetitions of the same starting structure probably isn't enough to conclude anything. And why is only one shown in Figure S10? Where is the data for the other two simulations?

What is the "jointing" region? Presumably the border between the PDGu and the PBLK segments – please use a different word in the main text and in the Supp info.

Overall, the data represents a small amount of superficial modelling to generate something that fits the author's expectations. They have placed the PBLK helix in water and saw it unfold, and put the PBLK helix on the membrane and saw it remain a helix. This is not necessarily "wrong" or "meaningless" - but the fact that it remains helical on the membrane and loses the secondary structure in water is a valid result, but it doesn't prove anything. If the authors could take the coiled structure from water and show it forms a helix on the membrane, that would be interesting. But that might not be achievable in a reasonable timeframe.

Fig 3e proposes quite specific structural interactions with no evidence. In particular, (II) is quite explicit about the assembly of the lipids and the helix – this is completely speculative and together with (III) and (IV) should be removed.

The Discussion requires significant revision:

Page 12, line 277. "The reduced tendency of the block copolymer to bind mammalian membranes is probably linked to their less negatively charged surface". As indicated above, structural change is not indicative of binding strength, this needs to be restated.

Page 12, line 286. "expose a sheath of cationic charges and a regular helical main chain hydrophobic inner core." This sentence is nonsensical. Firstly, what is a "sheath" and secondly the hydrophobic core is not exposed.

Page 12, line 287. "Computer simulation shows that the helix probably contributes towards strong

interactions with the anionic bacterial lipids". The simulation showed no evidence of strong interactions.

Page 13, line 293. "Our block copolymer possesses unique bacteria-triggered radial amphiphilicity (RA) in which the hydrophobic domain (backbone) is hidden in free solution to reduce hemolysis and toxicity, but the cationic charges are selectively regularized and concentrated whilst the hydrophobic backbone is more exposed by anionic bacterial lipids to then result in strong bacteria-copolymer interactions."

I am not enamoured with the term "radial amphiphilicity". However, peptide backbones are not considered the hydrophobic domain in amphipathic sequences. Even though the authors consider the beta-peptide backbone to be hydrophobic, once helical structure is adopted, the side chains dominate the interactive topology. Also, the term "selectively regularised and concentrated" makes no sense – the whole sentence needs to be revised.

Page 13, line 316-317. "The block copolymer also appears superior to commercial non- bactericidal biofilm dispersal agents such as dispersin B or DNase. The authors have not presented any data to substantiate this claim – delete.

In summary, while the peptide has interesting selective antibacterial properties, the authors have attempted to provide a structural mechanism for this activity but have over-interpreted experimental data leading to unsubstantiated conclusions.

Reply to Reviewers

Please find our point-by-point responses below, where reviewers' comments are in black, our responses are in red, and the changes made in the manuscript are in blue. (Other minor grammatical corrections we made ourselves are in green.)

Reviewer #1 (Remarks to the Author):

1) Apart from the MIC data in Table 1 it appears that most, if not all, the *in vitro* and *in vivo* studies were performed with a single MRSA strain USA300. Rather than test a few ill-characterized MRSA strains **it is necessary to test activity against well characterized MRSA strains representing the major lineages**. In particular, it is necessary to demonstrate activity against hospital-associated MRSA strains with resistance to multiple conventional antibiotics (including glycopeptides vancomycin and daptomycin) because these cause the majority of nosocomial device-related infections involving biofilm formation, as well as the majority of nosocomial bacteraemia/septicaemia.

Reply: We have now tested our copolymer against several (17) well-characterized MRSA strains representing the major lineages (new Table 3, which is reproduced on the next page). Our copolymer shows similar activity against MRSA USA300 as against the (17) hospital-associated MRSA strains which are resistant to multiple conventional antibiotics (including vancomycin and daptomycin) (new Table 3). The copolymer MICs against all the 17 new HA-MRSA strains tested (8-16 µg/mL) are comparable to that for MRSA USA300 (16 µg/mL); the 17 MRSA strains included 7 vancomycin-resistant *S. aureus* (Strains #1 to 7, Table 3), 7 daptomycin non-susceptible vancomycin-intermediate *S. aureus* (Strains #8 to 14), and 3 multi-drug resistant (MDR) MRSA strains (Strains #15 to 17).

The new data are added to the main text as new Table 3 (page 36). The following sentences have been added in the main text:

(new page 7, line 160-163)

“Further profiling demonstrated that the copolymer is also potent against other MRSA strains from major lineages of global epidemiology⁴⁸, including (HA-)MRSA strains resistant to multiple conventional antibiotics (including vancomycin, daptomycin) (Table 3).”

(new page 13, line 299-303)

“We also show that our copolymer is just as effective against HA-MRSA strains with resistance to multiple conventional antibiotics (Table 3, Fig. 5). Multi-drug resistant (MDR) HA-MRSA bacteria cause the majority of nosocomial bacteraemia/septicaemia and device-related infections involving biofilm formation.”

Table 3. Antimicrobial activity of **PDGu(7)-b-PBLK(13)** against multi-drug resistant clinically isolated MRSA.

	Serial No.	Designation	MIC (µg/mL)		Multi-drug resistance	Major lineage/ Clonal Complex ⁴⁸
			PDGu(7)-b-PBLK(13)	Resistant Antibiotic		
VAN resistant S. aureus	1	HIP11714	16	512	CIP, CLI, ERY, GEN, LVX, MXF, OXA, RIF, TEC	5
	2	HIP11983	16	16	CIP, CLI, ERY, GEN, LVX, MXF, OXA, TET	5
	3	HIP13170	16	128	CIP, CLI, ERY, GEN, LVX, MXF, OXA, TEC, TET	5
	4	HIP13419	16	VAN 64	CIP, CLI, ERY, GEN, LVX, MXF, OXA, TEC, TET	5
	5	HIP14300	16	32	CIP, CLI, ERY, LVX, MXF, OXA, TEC	5
	6	HIP15178	16	512	CIP, CLI, ERY, LVX, MXF, OXA, TEC	5
	7	AIS2006032	16	>512	CIP, CLI, ERY, LVX, MXF, OXA, TEC	5
DAP non-susceptible VAN intermediate S. aureus	8	HIP09433	16	4	CIP, ERY, GEN, LVX, MXF, OXA, PEN, TMP	45
	9	SAMER-S6	16	16	TMP, PEN, TEC	5
	10	6820	16	8	OXA, RIF, TEI	5
	11	TTSH-478700	8	DAP 16	CIP, LVX	22
	12	TTSH-671549	16	8	CIP, ERY, LVX	22
	13	TTSH-478701	8	4	CIP, ERY, LVX, RIF	22
	14	ATCC 700789	16	4	CIP, ERY, LVX, RIF, TOB	5
MDR MRSA	15	ATCC BAA38	16	128	PEN, STR	8
	16	ATCC BAA39	16	TET 128	CIP, ERY, GEN, IPM, LVX, PEN, TMP, TOB	8
	17	ATCC BAA44	16	32	CIP, ERY, GEN, LVX, PEN, TOB	8

CIP, ciprofloxacin; CLI, clindamycin; DAP, daptomycin; ERY, erythromycin; GEN, gentamicin; IPM, imipenem; LVX, levofloxacin; MXF, moxifloxacin; OXA, oxacillin; PEN, penicillin; RIF, rifampicin; STR, streptomycin; TEC, teicoplanin; TET, tetracycline; TMP, trimethoprim; TOB, tobramycin; VAN, vancomycin

Information on various new strains in the new Table 3 has been added in the Supplementary Information (page S26):

“Vancomycin-resistant *S. aureus* (Strains #1 to 7) were kindly provided and tested by Prof. Barry Kreiswirth and Dr. José Mediavilla of the Public Health Research Institute, University of Medicine and Dentistry of New Jersey (USA). Daptomycin non-susceptible vancomycin-intermediate MRSA (Strains #8 and 9) were kindly provided by BEIresources.org. Strain #10 was kindly provided by Dr. Adriana Rosato of the Houston Methodist Research Institute (USA). Strains #11 to 13 were kindly provided by Tan Tock Seng Hospital (TTSH, Singapore). Strains #14 to 17 were purchased from ATCC. The characterization information can be retrieved from BEIresources.org or ATCC.org. Multilocus sequence typing (MLST) characterization for the three VISA strains from the local hospital (Strains #11 to 13) were conducted following the protocols previously reported⁸ with minor modifications.”

2) *In vitro* biofilm formation. The methods and figure legend do not specify if these experiments were performed with USA300. If they were then the type of biofilm formed under the growth conditions employed (glucose containing broth) is likely to involve cell wall anchored proteins in the accumulation phase (see McCourt J, Fibronectin-binding proteins are required for biofilm formation by community-associated methicillin-resistant *Staphylococcus aureus* strain LAC. FEMS Microbiol Lett. 2014 Apr;353(2):157-64.....). While some HA-MRSA strains form CWA protein-dependent biofilm *in vitro* a significant proportion of MRSA and also *S. epidermidis* form biofilm that involves the polysaccharide intercellular adhesin encoded by the *ica* locus. Before the authors can claim that their molecule is effective against *S. aureus* biofilm this must be investigated further.

Reply: The relevant captions of Figures 2, 4 and 6 and methods (page 19-22) are now revised to include the descriptor “MRSA USA300”. Further, we have now investigated the efficacy of the copolymer against HA-MRSA and methicillin-resistant *S. epidermidis* (MRSE) biofilms that involve the polysaccharide intercellular adhesin encoded by the *ica* locus¹ (new Figure 5). We tested our copolymer against 6 new HA-MRSA/MRSE strains, specifically MRSA (a1) ATCC BAA38, (a2) ATCC BAA39, (a3) ATCC BAA40, and (a4) ATCC BAA44, and MRSE (b1) ATCC 35984, and (b2) ATCC 700563. Our copolymer was superior to vancomycin in eradicating the 6 biofilm bacteria tested. These results indicate that the biofilm eradication efficacy of our copolymer **PDGu(7)-b-PBLK(13)** is not limited to just MRSA USA300 biofilm which involves cell wall anchored proteins, but also applies to the major types of biofilm formed by MRSA and MRSE.

The new data have been updated in the main text as new Figure 5 (new page 32), shown below. The detailed protocol has been added in Supplementary Information Section 2.9 (page S50). The following paragraph has been added in the main text:

(new page 11, line 250-259):

“(CA-)MRSA USA300 maintained in a broth medium supplemented with glucose typically forms biofilm involving cell-wall anchored protein (fibronectin-binding proteins)⁵⁷, whilst many (HA-)MRSA^{58,59} and *S. epidermidis*^{60,61} strains form biofilms involving the polysaccharide intercellular adhesin encoded by the *ica* locus⁶². To determine if the block copolymer is active against other types of biofilms, biofilms formed by various HA-MRSA and methicillin-resistant *S. epidermidis* (MRSE) strains under conditions promoting the *ica* locus expression⁶³ were treated with the copolymer. Our copolymer **PDGu(7)-b-PBLK(13)** was more active than vancomycin in eradicating the biofilms of HA-MRSA and MRSE strains (Fig. 5). Hence, our copolymer is effective not only against MRSA biofilms involving fibronectin-binding protein, but also against other major types of biofilm formed by HA-MRSA and MRSE.”

Figure 5. PDGu(7)-b-PBLK(13) shows dose-dependent eradication of biofilm bacteria ((a) HA-MRSA and (b) MRSE) under conditions that promote polysaccharide intercellular adhesion; y-axis: biofilm bacteria (CFU/mm² peg) formed by HA-MRSA (a1) ATCC BAA38, (a2) ATCC BAA39, (a3) ATCC BAA40, (a4) ATCC BAA44 and MRSE (b1) ATCC 35984, (b2) ATCC 700563. (Vancomycin is used as antibiotic control.) Data are presented as mean ± standard deviation.

3) Daptomycin has been used to treat persistent bacteraemia as an alternative to vancomycin and indeed when vancomycin insensitive strains are involved. Daptomycin resistance involves changes in membrane lipid composition (gain of function *MprF* mutants) which result in an increase in lysyl-phosphatidyl glycerol and hence an increased positively charged membrane. Does the co-beta peptide have activity in mutants where membrane lipid composition is altered?

Reply: We have now tested our copolymer against daptomycin(DAP)-resistant MRSA strain **6820** (#10 in new Table 3) which is a clinical isolate with reported *MprF* mutation and reduced susceptibility to vancomycin and daptomycin (ref. 2 in text). (**6820** was kindly provided by Dr. Adriana Rosato of the Houston Methodist Research Institute (USA).) Our copolymer retains the same MIC against **6820** compared to the control strain *Staphylococcus aureus* (SA) ATCC 29213 which is pan-sensitive and daptomycin-susceptible (**Table R1-**

Q3a). On the other hand, the MICs of vancomycin and daptomycin against **6820** were increased by 4 folds and 8 folds, respectively, compared with those of the SA29213 control.

We also tested 3 DAP-resistant MRSA isolates from the local hospital (Strains #11-13, new Table 3). We also identified single-nucleotide polymorphism (SNP) mutations in the *MprF* gene of these DAP-resistant strains (#11-13). These *MprF* mutants show 4- to 16-fold increase in daptomycin MIC (compared to the daptomycin-susceptible control strain SA29213). However, our copolymer retained its MIC against these DAP-resistant clinical strains, further confirming its activity against *MprF* mutants.

Further, our copolymer retains its good efficacy against **6820** persister cells (**Table R1-Q3b**): it can remove more than 99.9% of bacterial burden (defined as MBC, minimal bactericidal concentration) at 1×MIC while daptomycin and vancomycin even at 50×MIC cannot remove a similar proportion of the persister bacterial cells. (Nutrient-deprived persister cells were generated using the protocol described in Supplementary Information page S32).

Table R1-Q3 (a) Activity of copolymer and antibiotic controls (vancomycin and daptomycin) against replicating DAP-resistant MRSA strains (with *MprF* mutations).

Strain No.	Mutant Designation	MIC _{mutant} /MIC _{control}		
		PDGu(7)-b-PBLK(13)	VAN	DAP
10	6820	1	4	8
11	TTSH-478700	0.5	8	16
12	TTSH-671549	1	4	8
13	TTSH-478701	0.5	4	4

†Control strain is SA29213; MIC is minimum inhibitory concentration.

Table R1-Q3 (b) Activity of copolymer and antibiotic controls (vancomycin and daptomycin) against **6820** persister cells.

Strain No.	Mutant Designation	MBC _{mutant} /MIC _{control}		
		PDGu(7)-b-PBLK(13)	VAN	DAP
10	6820	1	>50	>50

†MBC is minimum bactericidal concentration

4) Expression of b-lactam resistance in MRSA requires the integrity of the membrane lipid rafts enriched with staphyloxanthin and containing the membrane associated protein flotillin. Any disruption to the integrity of lipid rafts (and/or PBP2a) results in bacteria becoming susceptible to beta-lactams. There is considerable interest in resurrecting b-lactams to combat MRSA with molecules that act synergistically. Does the co-polymer have synergistic activity with b-lactams against MRSA?

Reply: We have now investigated the possible synergy effect of our copolymer with various beta-lactam antibiotics against MRSA USA300 using the checkerboard method; from the fractional inhibitory concentration (FIC) values, we found that combinations of our

copolymer with beta-lactam antibiotics show only partial to no synergy (but additive) effects. The results are summarized in **Table R1-Q4** below.

(Synergy is defined using the FIC values³: if $FIC < 0.5$, the combination is synergistic; if $0.5 \leq FIC < 1$, there is partial synergy; if $FIC = 1$, there is only additive effect but no synergy; if $(2 \leq FIC < 4)$, the combination is indifferent; if $(FIC > 4)$, the combination is antagonistic.)

Table R1-Q4. Checkerboard synergy test of copolymer and beta-lactam antibiotics against MRSA USA300.

Antibiotics	MIC of antibiotic alone (µg/mL)	MIC of antibiotic in combination (µg/mL)			FIC	Synergy?
		+1/2 MIC copolymer	+1/4 MIC copolymer	+1/8 MIC copolymer		
Ampicillin	32	8	32	32	0.75	Partial
Carbenicillin	32	8	16	16	0.75	Partial
Ceftazidime	128	64	64	128	0.75	Partial
Penicillin	1	0.25	0.5	0.5	0.75	Partial
Oxacillin	1	1	1	1	1	No [†]
Piperacillin	16	8	16	16	1	No [†]

[†] Only additive effect

5) Table 1 describes MIC of a “panel of Gram-positive bacteria” but apart from *S. aureus* only *Bacillus subtilis* is shown. Does the molecule have activity against other Gram-positive pathogens such as *Streptococcus pneumoniae* and other *Streptococci*. There is no mention of activity against Gram-negative bacteria. Can the authors provide information on any activity against ESKAPE pathogens?

Reply: We have now performed additional tests against other Gram-positive (including *Streptococcus pneumoniae* and other *Streptococci*) and ESKAPE pathogens. The copolymer is active against the Gram-positive *Bacillus cereus*, *Listeria monocytogenes* and *Staphylococcus epidermidis* at concentrations comparable to that for *S. aureus* (Strains #1-4, **Table R1-Q5**). However, the copolymer is 4 to 16 times less potent against *Enterococci* and *Streptococci* (Strains #5-11, **Table R1-Q5**), and is 8 to 32 times less potent against the Gram-negative bacteria tested (Strains #12-15, **Table R1-Q5**).

On the other hand, our copolymer can disperse the biofilms of all Gram-positive bacteria we tested (including *Streptococcus agalactiae*, *Streptococcus pyogenes*, *Streptococcus pneumoniae*, *Streptococcus mutans* and *Enterococcus faecalis*). (**Figure R1-Q5**).

Table R1-Q5. MIC values of copolymer against ESKAPE pathogens and other clinically relevant Gram-positive bacteria.

Strain No.	Bacteria strains (Gram-positive)	Designation	Copolymer MIC ($\mu\text{g/mL}$)
1	Bacillus cereus	ATCC 11778	16
2	Listeria monocytogenes	ATCC 19115	16
3	Staphylococcus epidermidis	ATCC 35984	8
4	Staphylococcus epidermidis	ATCC 700563	8
5	Enterococcus faecium	ATCC 29212	128
6	Enterococcus faecalis	VRE V583	256
7	Streptococcus agalactiae	ATCC 13813	64
8	Streptococcus iniae	ATCC 29178	128
9	Streptococcus parasanguinis	ATCC 15912	64
10	Streptococcus pneumoniae	ATCC BAA334	256
11	Streptococcus pyogenes	ATCC 12344	64
Bacteria strains (Gram-negative)			
12	Acinetobacter baumannii	ATCC 19606	256
13	Escherichia coli	ATCC 25922	128
14	Klebsiella pneumoniae	ATCC 12993	512
15	Pseudomonas aeruginosa	ATCC 27853	512

Figure R1-Q5. PDGu(7)-b-PBLK(13) (purple) shows dispersal of biofilm biomass of different Gram-positive bacteria: **(a)** *Streptococcus agalactiae* ATCC 13813, **(b)** *Streptococcus pyogenes* 12344, **(c)** *Streptococcus pneumonia* 6303, **(d)** *Streptococcus mutans* 10449 and **(e)** *Enterococcus faecalis* V583. The control antibiotic/antiseptic control (grey) is ineffective in the biomass removal. Biomass is quantified by crystal violet staining.

6) Development of resistance. A large inoculum of *S. aureus* cells $>10^9$ were exposed to $10\times$ MIC for a prolonged period. What happened if bacteria were exposed to the MIC or below the MIC with surviving populations being exposed to slightly increased concentrations? This approach would allow small incremental increases which would mimic better the situation *in vivo*. For example VISA strains emerge after acquisition of 5 or 6 mutations in different chromosomal genes.

Reply: We conducted another assay to attempt to evolve resistance in MRSA by daily serial passaging in the presence of sub-inhibitory concentrations of copolymer or antibiotic (control). During the 14-day test, no resistance to the copolymer was observed (with only a 2-fold increase in MIC) (new Supplementary Figure S10). Resistance was not observed in any of the colonies tested on the 14th day. Conversely, resistance to the antibiotic control (ciprofloxacin) was rapidly selected for. The test was done with three independent biological replicates. It seems that the copolymer mutation frequency is much lower than that of the classical antibiotic ciprofloxacin even though we cannot be completely certain that selection of escape mutants resistant to the copolymer is not possible. It is possible that emergence of resistance requires several independent mutations, which could not be observed under our experimental design.

The detailed protocol has been added in the new Supplementary Section 2.5 (page S34) and the new data have been updated in the manuscript as new Supplementary Figure S10 (page S34) and shown below. The following sentences have been added in the main text:

(new page 7, line 167-171)

“We then tried to select mutants by the continued pressure of sub-inhibitory concentrations of the block copolymer for up to 14 days (as described before⁵¹). This approach also did not select for copolymer-resistant MRSA colonies. As a control, escape mutants resistant to the antibiotic ciprofloxacin were easily selected (Supplementary Fig. S10).”

Figure S10. Resistance development of MRSA USA300 by serial passage with sub-inhibitory concentrations of the agent. Data are presented as folds MIC change over time. **(a)** PDGu(7)-b-PBLK(13) and **(b)** Ciprofloxacin antibiotic control. All tests were done with three biological replicates.

Reviewer #2 (Remarks to the Author):

This paper describes the synthesis and antimicrobial activity of a novel block co-polymer containing D-glucose and beta lysine. This block polymer was designed to be amphipathic, containing negatively charged glucose side chains at the N-terminus and positively charged residues at the C-terminal end. In this way, it was hypothesised that the glucose side chains would facilitate penetration into the peptidoglycan wall and the lysine residues would mediate binding and disruption of the inner phospholipid membrane bilayer.

A significant amount of work has been presented. The results demonstrate good antibacterial activity and low host toxicity which is a challenge to achieve and demonstrates the potential of these materials. However, there are a number of issues which I feel diminish the novelty and quality of the studies and which need to be addressed before this manuscript can be considered for publication. **In particular, there are significant issues with the CD analysis, the computer simulation and referencing of prior work.**

1) Page 4, Line 69, the authors state “beta-peptides have relatively hydrophobic backbones”. While beta peptides have an extra methylene in the backbone and therefore additional hydrophobicity related to these addition atoms, the relative hydrophobicity of beta peptides is more controlled by the structure of the side chains. This statement should be re-stated.

Reply: The statement has been revised in the main text to the following:

New page 4, line 73-74:

“Compared to alpha-peptides, beta-peptides have an extra methylene group in the backbone. The hydrophobicity of beta-peptides may be tuned by the structure of the side chains.”

Old page 4, line 69:

“Compared to alpha-peptides, beta-peptides have relatively hydrophobic backbones and tend to form radially amphiphilic (RA) structures in which the hydrophobic backbones are hidden which may show reduced hemolysis and toxicity.”

The authors need to take greater care in accurate citation:

2) old Page 3, line 66. Old References 23-25 are incorrect. Ref 23 is a cyclic alpha-peptide with a beta hairpin structure. Ref 24 is a review on beta-peptoids and the authors should refer to primary papers as specific examples. Ref 25 is a large review on the biochemistry and medicinal chemistry of the dengue virus protease – again the authors should cite the specific paper relevant to their paper.

Reply: Thanks for your comments. Old Ref. 23 is now changed to (new) Ref. 28 (Porter, E. A., Weisblum, B. & Gellman, S. H. *Journal of the American Chemical Society* **124**, 7324-7330 (2002)) which studied the impact of cationicity, hydrophobicity and helical propensity on antimicrobial and hemolytic activities of a series of beta peptides.

Old Ref. 24 is now changed to (new) Ref. 29 (Liu, D. & DeGrado, W. F. *Journal of the American Chemical Society* **123**, 7553-7559 (2001)) which specifically reports the effect of hydrophobicity on microbial selectivity of antimicrobial beta-peptides using model membrane systems.

Old Ref. 25 is now changed to (new) Ref. 30 (Eband, R. F., Raguse, L., Gellman, S. H. & Eband, R. M. *Biochemistry* **43**, 9527-9535 (2004)) which specifically studies the membrane-active properties of antimicrobial beta-peptides in detail.

3) Page 5. The authors need to refer to much earlier studies which reported the synthesis of nylon-3 and analogues together with structural analysis. These papers are from the group Munoz-Guerra, Subirana *et al.* and include (amongst at least 8) *Nature*, 1984, 311, 53-54; *Macromolecules* 1998, 31, 124-134; *Macromolecules* 1987, 20, 62-68.

Reply: The mentioned publications have been cited accordingly as new Ref. 41-43. The following sentence has been added to the main text:

(new page 4, line 77-78)

“Munoz-Guerra and Subirana *et al.* reported the first research on nylon-3 and analogues, which included the synthesis and helical propensity of these beta-peptides⁴¹⁻⁴³.”

4) The authors need to state how their work differs and improves on these earlier studies. The review by Konai *et al.* [*Biomacromolecules* (2018) 19, 1888–1917] provides a very detailed overview of the application of polymers with a section on nylon-3/poly-beta lactams. Again, how does the chemistry and design presented by the authors differ and/or improve on what has been previously published?

Reply: Our work differs from these earlier works in the following ways, and these differences have been updated in the revised main text:

- a) **(new page 4, line 78-84):** In the development of antimicrobial beta-peptides, previous efforts focus mainly on random co-beta-peptides and optimization of their cationic *versus* hydrophobic beta-lactam residues to reduce hemolysis whilst maintaining good bactericidal effect⁴⁴⁻⁴⁷. There is no reported work on glycosylated block co-beta-peptides. Block co-beta-peptides are interesting as they may show unique combinations of properties displayed by the individual blocks which are as yet under-exploited for the development of next-generation antibacterials.
- b) **(new page 4, line 84-88):** Also, a strategy for the facile synthesis of block co-beta-peptides has not been previously reported. In this study, we report a simple one-shot one-pot anionic ring opening (co)polymerization (AROP) strategy to synthesize a new series of enantiomeric block co-beta-peptides which cannot be made by sequential copolymerization.
- c) **(new page 4, line 92-99):** Further, the block co-beta-peptide has interesting biological properties. Unlike classical antibiotics, **PDGu(7)-b-PBLK(13)** retains potency against **MRSA persister cells** and biofilms. It is active against both the community-acquired (CA-) and hospital-associated (HA-) MRSA strains. The block copolymer also effectively removes biofilm biomass **but the homocationic beta-peptide (PBLK(20)) cannot**. The block copolymer is bactericidal against MRSA in various murine models of systemic acute and established infections, and also in an *ex vivo* human skin infection model, while having no *in vivo* acute toxicity in murine repeated dosing studies.

5) The key papers of the Gellman group should also be cited: Porter EA, Weisblum B, Gellman SH. *J Am Chem Soc.* 2002 Jun 26;124(25):7324-30. Porter EA, Wang X, Lee HS, Weisblum B, Gellman SH. *Nature.* 2000 Apr 6;404(6778):565.

Reply: The mentioned key publications of the Gellman group are now cited as (new) Ref. 23 and 28 in the revised main text:

New page 3, line 66:

“Amongst the various synthetic polymer families being explored as peptidomimetics²¹⁻²⁵, beta-peptides are promising...”

New page 3, line 69:

“Beta-peptides have been considered for use in diverse therapeutic applications such as antimicrobial agents²⁸⁻³⁰...”

A few follow-up studies from Gellman group have also been cited as Ref. 44-47 in the revised main text:

(new page 4, line 78-81)

“In the development of antimicrobial beta-peptides, previous efforts focus mainly on random co-beta-peptides and optimization of their cationic *versus* hydrophobic beta-lactam residues to reduce hemolysis whilst maintaining good bactericidal effect⁴⁴⁻⁴⁷...”

Ref. 23: Porter, E. A., Wang, X., Lee, H.-S., Weisblum, B. & Gellman, S. H. Antibiotics: Non-haemolytic β -amino-acid oligomers. *Nature* **404**, 565 (2000).

Ref. 28: Porter, E. A., Weisblum, B. & Gellman, S. H. Mimicry of host-defense peptides by unnatural oligomers: antimicrobial β -peptides. *Journal of the American Chemical Society* **124**, 7324-7330 (2002).

Ref. 44: Mowery, B. P. *et al.* Mimicry of antimicrobial host-defense peptides by random copolymers. *Journal of the American Chemical Society* **129**, 15474-15476 (2007).

Ref. 45: Mowery, B. P., Lindner, A. H., Weisblum, B., Stahl, S. S. & Gellman, S. H. Structure–activity relationships among random nylon-3 copolymers that mimic antibacterial host-defense peptides. *Journal of the American Chemical Society* **131**, 9735-9745 (2009).

Ref. 46: Liu, R. *et al.* Structure–activity relationships among antifungal nylon-3 polymers: identification of materials active against drug-resistant strains of *Candida albicans*. *Journal of the American Chemical Society* **136**, 4333-4342 (2014).

Ref. 47: Liu, R. *et al.* Tuning the biological activity profile of antibacterial polymers via subunit substitution pattern. *Journal of the American Chemical Society* **136**, 4410-4418 (2014).

6) Page 5, line 104. The authors state “The molecular weight of the product increased monotonically..”. Do the authors mean linearly?

Reply: We have now revised the text to the following:

(new page 5, line 109)

“The molecular weight of the product increased **linearly** over an 8-hour period...”

7) Page 5. The authors refer to estimated molecular weights based on the GPC retention. Why did they not use mass spectrometry to determine accurate mass?

Reply: We have now measured the polymer molecular weights by Matrix-Assisted Laser Desorption/Ionization-Time of Flight (MALDI-TOF) mass spectrometry. We added the following sentences to the main text:

(new page 6, line 130-134)

“The molecular weights (M_n) of the homocationic **PBLK(20)**, homosugar **PDGu(20)** and copolymer **PDGu(7)-*b*-PBLK(13)** were respectively 3012 Da, 3159 Da and 3391 Da, as measured by Matrix Assisted Laser Desorption/Ionization-Time of Flight (MALDI-TOF) mass spectroscopy (Supplementary Fig. S6).”

In the Supplementary Information pages S24-25, we added the protocols and the new MALDI-TOF data:

Figure S6. MALDI-TOF analysis of (a) homocationic **PBLK(20)**, (b) homosugar **PDGu(20)** and (c) block copolymer **PDGu(7)-*b*-PBLK(13)**.

8) Page 6. If NMR etc suggests a ratio of G7:K13, why do the authors persist in referring to the product as 10:10 – this was the target ratio – but using this nomenclature throughout the manuscript gives the wrong impression that this is indeed the precise composition.

Reply: The nomenclatures of the block copolymers have now been corrected to the actual composition. For example, **PDGu(10)-b-PBLK(10)** is now changed to **PDGu(7)-b-PBLK(13)** throughout the manuscript. Also, a new Table 1 has been added in the main text (new page 35) to summarise the design and actual compositions of all the copolymers.

Table 1. Design and actual ratios of **DGu** to **BLK** in **PDGu(x)-b-PBLK(y)** before and after deprotection.

Sample	Design ratio of DGu _p to BLK _p	Actual ratio ^a of DGu _p to BLK _p	Actual ratio ^b of DGu to BLK after deprotection
P1	PDGu _p (6.7)-b-PBLK _p (13.3)	PDGu _p (6)-b-PBLK _p (14)	PDGu(5)-b-PBLK(15)
P2	PDGu _p (8)-b-PBLK _p (12)	PDGu _p (8)-b-PBLK _p (12)	PDGu(6)-b-PBLK(14)
P3	PDGu _p (10)-b-PBLK _p (10)	PDGu _p (10)-b-PBLK _p (10)	PDGu(7)-b-PBLK(13)
P4	PDGu _p (12)-b-PBLK _p (8)	PDGu _p (12)-b-PBLK _p (8)	PDGu(9)-b-PBLK(11)
P5	PDGu _p (13.3)-b-PBLK _p (6.7)	PDGu _p (14)-b-PBLK _p (6)	PDGu(10)-b-PBLK(10)
P6	PDGu _p (20)	PDGu _p (20)	PDGu(20)

^a Ratios were calculated based on ¹H NMR integrations of (protected) PDGu_p(x)-b-PBLK_p(y)

^b Ratios were calculated based on ¹H NMR integrations of (deprotected) PDGu(x)-b-PBLK(y)

9) Page 6, line 120. “corroborating their blocky rather than random structures”. Should be “block”, not “blocky”. Also on line 140.

Reply: The instances of “blocky” have been corrected to “block” accordingly in the main text as indicated below:

New page 6, line 123-126:

“NMR spectroscopy measurements of **PDGu(x)-b-PBLK(y)** show two sets of signals belonging to **PDGu** and **PBLK** respectively, corroborating their **block** rather than random structures (Supplementary Fig. S3-S5).”

New page 7, line 148-149:

“Unexpectedly, with one-shot AROP with simultaneous feed of the two beta-lactams, we could achieve successful synthesis of the new family of **block** copolymers of **PDGu_p-b-PBLK_p**.”

New page 25, line 569-571:

“(c to e) Kinetic studies and (f) GPC measurements verify the well-controlled single chain **block** architecture of **PDGu_p(x)-b-PBLK_p(y)**.”

10) The antibacterial activity appears promising – why was the vancomycin dose of 2.5mg/kg used in the *in vivo* studies? This needs to be explained as the authors claim that their peptide is superior to vancomycin.

Reply: We apologize for this imprecision. We meant to say that vancomycin was used as the antibiotic control at the same dosing (at 2.5mg/kg × 6 times over 2 days) as our copolymer; the latter outperforms the clinically used vancomycin (control). The word “vancomycin” in Figure 6 and its caption has been replaced by “vancomycin control”.

New page 34, line 629-632:

“**Figure 6(f)** ... “*In vivo* antimicrobial activity of **PDGu(7)-b-PBLK(13)** against MRSA USA300 in an established murine excision wound model. Vehicle alone (–), **PDGu(7)-b-PBLK(13)** or vancomycin **control at the same dosing** (*i.e.* 2.5 mg/kg) were applied six times over 2 days, starting 72-hours post infection.”

11) The use of the word “probably” is inappropriate – the authors need to be more circumspect in their conclusions – 3 examples are listed below:

Page 8, line 175. “copolymer with its hydrophilic sugar block probably aggregates at the membrane interface, leading to the observed larger periplasmic gap.”

Page 9, line 199, “The accumulation of the block co-beta-peptide at the outer leaflet of the cytoplasmic membrane probably causes the increased periplasmic space visible in the cryo-TEM (Fig. 2d),”

Page 9, line 209, “copolymer disturbs the cell envelope which includes probably the membrane, the membrane-..”

Reply: The word “probably” has been removed from the manuscript in the various places:

New page 8, line 190-191 has been changed to:

“The copolymer with its hydrophilic sugar block ~~probably~~ aggregates at the membrane interface, leading to the observed larger periplasmic gap.”

New page 9, line 214-215 has been changed to:

“The accumulation of the block co-beta-peptide at the outer leaflet of the cytoplasmic membrane ~~probably~~ causes the increased periplasmic space visible in the cryo-TEM (Fig. 2d)”

New page 10, line 223-225 has been changed to:

“Taken together, the copolymer disturbs the cell envelope which includes ~~probably~~ the membrane, the membrane-cell wall interface, and also the cell wall.”

New page 13, line 303-307 has been changed to:

“The ability of our co-beta-peptide to kill all the sub-populations (planktonic, persister and biofilm states) of MRSA bacteria is ~~probably~~ attributable to mechanism(s) of kill -- membrane disruption and interface weakening effects which are not related to metabolism. The reduced tendency of the block copolymer to bind mammalian membranes is ~~probably~~ linked to their less negatively charged surface^{36,67}.”

There are a number of issues with the CD analysis:

The CD spectra shown in Fig 3c and 3d show the structure in water and POPG respectively of PDGu(20), PBLK(20) and the 10:10 co-block peptide. Fig 3c and d indicates that both

PDGu(20) and the 10:10 peptide adopt secondary structure in water and in POPG, but PBLK only adopts structure in POPG liposomes.

In addition, Fig S10 e, f and g show the CD spectra of these 3 peptides at 3 different pHs, water and POPG and POPC. This data further demonstrates the persistence of the helical structure for PDGu(20) in all conditions while PBLK(20) only adopts structure in both lipids. The block copolymer adopts structure in all solutions, indeed, the spectra appear to be composite in shape of the spectra in Fig S10 e and f.

12) Based on spectra published on other beta-peptides, the authors conclude the presence of a left-handed helix as beta-peptides containing non-cyclic beta amino acids adopt helical structures which are left-handed in conformation. However, beta-peptides containing cyclic beta-amino acids adopt different helical structures to beta peptides containing non-cyclic amino acids. This is quite evident when one compares the spectral minima for PDGu(20) [~222nm] and PBLK(20) [208-210nm]. The authors need to take great care to not over-interpret the spectra.

Reply: We agree with this insight, and added the following sentences into the main text:

(new page 9, line 208-211):

“It is known that beta peptides containing cyclic beta-amino acids adopt different helical structures to those containing non-cyclic amino acids³⁹. From the CD spectroscopy data, we see that (cyclic) PDGu spectral exhibited minima at 220nm while the (non-cyclic) PBLK spectral exhibited minima at 213nm (Fig. 3d).”

13) In addition, the authors state on page S38 that PBLK(20) interacts strongly with anionic lipids because of the change in spectra from an extended coil in the absence of lipid to a helical structure in POPG. However spectral changes do not provide any information on binding affinity, only partial information on structural changes between different solutions. This statement needs to be changed.

Reply: The sentence has been revised as below to delete the part on correlation of CD with binding affinity:

New Supplementary Information page S45:

“Circular dichroism study shows that the homocationic (**PBLK(20)**) transitions from a random coil to a left-handed helix.”

Old page S38:

“Circular dichroism study shows that the homocationic (**PBLK(20)**) ~~interacts strongly with anionic lipids to cause~~ transitions from a random coil to a left-handed helix.”

14) a) Moreover, spectra were obtained at a peptide concentration of 0.05 mg/ml (and re-stated as 50µg/ml elsewhere – need to be consistent) – **b)** given that the mass of each peptide is different, the changes in spectral intensity cannot be directly related to degree of secondary structure between peptides in any solution as molar ellipticity has not been calculated.

a) Reply: The peptide concentration has now been re-stated as 0.05mg/mL throughout the main text as below:

(new page 22, line 504-506)

“Polymers were dissolved at 0.05 mg/mL in different media *i.e.* DI water, 10 mM phosphate buffer (pH 2.6-8.7), 20 mM carbonate buffer (pH 10.8) and in the presence of 1mg/mL POPG or POPC liposomes.”

b) Reply: The molar ellipticity values $[\theta]$ of the polymers (*i.e.* PDGu(20), PBLK(20), and PDGu(7)-b-PBLK(13)) are now calculated based on their respective molecular weights as measured by MALDI-TOF (*i.e.* 3159 Da, 3012 Da and 3391 Da respectively), and the new molar ellipticity CD spectra are presented in new Fig. 3c,d and Supplementary Fig. S12l-n (see below too). Comparing the molar ellipticity curves of the 3 polymers in water, as well as in the presence of anionic POPG liposomes, we see that the secondary structure of the block copolymer (PDGu(7)-b-PBLK(13)) is the superimposition of the homosugar and homocationic structures.

Figure 3. (c, d) Molar ellipticity $[\theta]$ circular dichroism spectra of PDGu(20) (blue), PBLK(20) (red) and PDGu(7)-b-PBLK(13) (purple) in (c) DI and (d) in the presence of anionic POPG liposomes.

Figure S12. (l to n) Molar ellipticity $[\theta]$ circular dichroism spectra of (l) PDGu(20), (m) PBLK(20) and (n) PDGu(7)-b-PBLK(13) at 0.05 mg/mL in different media *i.e.* DI water, 10 mM phosphate buffer (pH 2.6-8.7), 20 mM carbonate buffer (pH 10.8) and in the presence of POPG or POPC liposomes.

15) a) What is the peptide:lipid (P:L) ratio in the CD experiments with liposomes? This is an important parameter to analyse the results especially since one of the main conclusions is that the lipids induce structure. **b)** However, the P:L ratio also influences the structure change.

a) Reply: Our CD spectra were obtained with peptide:lipid molar ratio of 1:79 to 1:89 (*i.e.* 0.05mg/mL peptide: 1mg/mL POPG lipid); this info is now added in the main text as below:

(new page 22, line 504-507)

“Polymers were dissolved at 0.05 mg/mL in different media *i.e.* DI water, 10 mM phosphate buffer (pH 2.6-8.7), 20 mM carbonate buffer (pH 10.8) and in the presence of 1 mg/mL POPG or POPC liposomes. (For POPG liposomes, the polymer:lipid (P:L) molar ratios of **PBLK(20)**, **PDGu(20)** and **PDGu(7)-b-PBLK(13)** are 1:79, 1:83 and 1:89 respectively).”

b) Reply: We further conducted the CD measurements at low (5:1) to high (around 1:80) P:L ratios while keeping the peptide concentration constant (**Figure R2-Q15**). We found that the CD signal increases as the P:L ratio increases but it plateaus at the charge neutralization ratio and beyond; the charge neutralization P:L ratios for the homocationic **PBLK(20)** and the copolymer **PDGu(7)-b-PBLK(13)** are 1:20 and 1:13 respectively. Hence our chosen P:L ratios of 1:79 and 1:89 are beyond the saturation ratio for both charged polymers.

Figure R2-Q15. Molar ellipticity $[\theta]$ circular dichroism spectra of (a) **PDGu(20)**, (b) **PBLK(20)** and (c) **PDGu(7)-b-PBLK(13)** in the presence of anionic POPG liposome at different P:L ratios.

There are a number of issues with the modelling work:

16) Choice of parameters. The authors state that they use the CHARMM general force field to model the beta peptides. Parameters for beta peptides are available in CHARMM, but if they used those parameters, they should cite Rathore *et al.* 2006 (doi: 10.1529/biophysj.106.084491). If they indeed just used the general CHARMM force field, they need to provide more details and justification. Particularly regarding the treatment of the backbone.

Reply: The details and justifications of our simulation model are now listed below and they have been updated in the revised Supplementary Information Section 2.8 (pages S41-42):

a) Rathore *et al.* developed parameters for beta-peptide modelling based on the similar chemical structures available in standard CHARMM force field²¹. The 1-4 Coulomb interactions of the beta-peptide in their simulation is scaled down by a factor of 0.4. However, this scaled-down factor is not compatible with the lipid model used in our simulation system, which is described by the CHARMM 36 lipid force field with a scaling factor of 1. Thus we didn't use their parameters for compatibility reasons.

- b) In our simulation, the beta-peptide parameters are created based on similar structures that are available in the CHARMM general force field. The backbone dihedral angles ϕ , θ , and ψ of the beta-peptide, as depicted by a typical beta-peptide in Figure S12a, are described by CG2O1-NG2S1-CG311-CG321, NG2S1-CG311-CG321-CG2O1 and CG311-CG321-CG2O1-NG2S1 in the CHARMM general force field, respectively, where the CG2O1 is the carbonyl carbon, NG2S1 is the peptide nitrogen, CG311 is the aliphatic carbon in CH group, and CG321 is the aliphatic carbon in CH₂ group. These parameters, according to the annotation from the force field, are obtained from the alanine dipeptide parameters, alkane parameters, and alanine dipeptide parameters, respectively.

Figure S12 (a) Definition of the three backbone dihedral angles in a typical beta-peptide.

- c) To further confirm the validity of our parameters, additional benchmark simulations were performed with two reported beta-peptide sequences, whose helical structures had already been proven. They are (i) sequence 1 (seq1): β -Glu- β -Leu- β -homoornithine- β -Phe- β -Leu- β -Asp- β -Phe- β -Leu- β -homoornithine- β -homoornithine- β -Leu- β -Asp²², and (ii) sequence 2 (seq2): ACHC- β -Lys- β -Leu-ACHC- β -Lys- β -Leu-ACHC- β -Lys- β -Leu²¹. Parameters of both sequences were developed with the same strategy as used in the present study. Both simulations started from the helical structures, and the stability of the helices were monitored during 100 ns simulation in water. As depicted in Figure S12b which shows the time evolution of the Root Mean Square Deviation (RMSD) of the peptide backbone atoms from the initial helical state (all less than 0.2 nm), both sequences are maintained as helices, suggesting that our parameters are able to characterize the helical structure of these beta-peptides.

Ref. 21: Rathore, N., Gellman, S. H. & de Pablo, J. J. Thermodynamic stability of beta-peptide helices and the role of cyclic residues. *Biophys J* **91**, 3425-3435 (2006).

Ref. 22: Daniels, D. S., Petersson, E. J., Qiu, J. X. & Schepartz, A. High-Resolution Structure of a beta-Peptide Bundle. *J Am Chem Soc* **129**, 1532-1533 (2007).

Figure S12 (b). Two additional benchmark simulations to confirm the validity of the force field parameters for beta-peptide. The helical structures of the testing sequences are well

maintained as indicated by the small RMS deviations of their backbones from the initial helical structure.

17) The composition of the peptides used for simulations – any conclusions based on modelling of either block of 10 or a 10:10 co-block is a rough approximation without accurate mass or structure.

Reply: We used the actual copolymer composition **PDGu(7)-b-PBLK(13)** in our simulation, and so our composition accurately reflects the actual structure of the copolymer. The following sentence has been added to new Supplementary Information page S45:

“A helical model of **PDGu(7)-b-PBLK(13)** was simulated.....This helix model was a combination of two well-defined helices, in which the **PBLK** block (13 units) was an L14 helix and the **PDGu** block (7 units) was also a left-handed helix obtained from a short simulation of an optimized structure in gas phase.”

18) Model membrane – which phospholipids were used for creating the model bilayer? There are no details apart from a comment about a negative lipid. Again, the results are not valid.

Reply: Sorry for the previous omission. The details of membrane model in our simulation are now added in the new Supplementary Information page S43 as below:

“The membrane model employed, which mimics the cytoplasmic cell membrane of *Staphylococcus aureus*, is a mixture of 1,1'-palmitoyl-2,2'-vacenoyl cardiolipin (PVCL2) and 3-palmitoyl-2-oleoyl-D-glycero-1-phosphatidylglycerol (POPG) in the molar ratio of 42:58, as described in Epanand and Epanand²³.”

Ref. 23: Epanand, R. M. & Epanand, R. F. Bacterial membrane lipids in the action of antimicrobial agents. *J Pept Sci* **17**, 298-305, doi:10.1002/psc.1319 (2011).

19) Distance between beta peptide and membrane. What exactly is the distance in Figure S10a? Center of mass? How far from the membrane was the peptide at the start of the simulation? And in what orientation?

Reply: The distance in new Figure S12i (old Figure S10a) refers to the distance between centers of masses of peptide and membrane and the z-distance is its z-component. A schematic illustration of the distance has now been added as new Supplementary Figure S12h (see below). The description has been updated in the caption of Figure S12h as shown below (page S48):

Figure S12 (h) Left panel: The initial configuration of the beta-peptide & membrane system. The membrane lies in the xy plane, and the peptide is placed parallel to the membrane surface with the z-component of distance between their centers of masses to be around 4.5 nm. Right panel: The bound configuration of the beta-peptide & membrane system. The z-component of the distance between the centers of masses decreases to around 2.5 nm.

The following descriptions are added in the Supplementary Information Section S2.8 (page S45):

“Initially, the **PDGu(7)-b-PBLK(13)** was placed above the membrane with the z-component of the distance between the centers of masses set to be around 4.5 nm; the central axis of the helix was placed parallel to the y axis for repeats 1 to 3, x axis for repeat 4 or diagonal of the xy plane for repeat 5 (Figure S12h shows one of the initial configurations).”

20) Sampling. Three repetitions of the same starting structure probably isn’t enough to conclude anything. And why is only one shown in Figure S10? Where is the data for the other two simulations?

Reply: We have now performed 5 repeats of the same starting helix structure. Three repeats were performed with same initial configuration but randomly assigned initial velocities. The data of all 3 repeats are now added to Figure S12i as repeats 1-3 (page S47) (also see below). Furthermore, we performed two new simulations of **PDGu(7)-b-PBLK(13)** with the membrane system using different starting configurations (repeats 4 and 5, Figure S12i): for the 4th and 5th simulation, the helix was placed at the same height with respect to the membrane as described above, but with different orientation of the helix central axis. For repeats 1-3, the peptide was set parallel to the y axis. For repeat 4, the orientation was set parallel to x axis, and for repeat 5, the peptide orientation was set as the diagonal of the xy plane. The peptides were found to adsorb onto the membrane surface in all the repeats.

Figure S12 (i) The Z-Distance between each of the blocks of the copolymer and the membrane. Z-distance refers to the z-component of the distance between the center of masses of the two objects (copolymer block and membrane).

21) What is the “jointing” region? Presumably the border between the PDGu and the PBLK segments – please use a different word in the main text and in the Supp info.

Reply: The “jointing region” was changed throughout the manuscript into “the border between the PDGu and the PBLK Segments” as below (new Supplementary Information page S45):

“In the border between the PDGu and the PBLK segments, the DGU units form H-bonds with the carbonyl groups of the BLK units, using their –OH groups on C3 as the H-bond donors (Figure S12i).”

The graphic representation is updated in Supplementary Figure S12j (page S47) to indicate the border between the 2 blocks as below:

Figure S12 (j) H-bond formation in the border between the PDGu and the PBLK blocks of the PDGu(7)-*b*-PBLK(13).

22) Overall, the data represents a small amount of superficial modelling to generate something that fits the author's expectations. **a)** They have placed the PBLK helix in water and saw it unfold, and put the PBLK helix on the membrane and saw it remain a helix. This is not necessarily “wrong” or “meaningless” - but the fact that it remains helical on the membrane and loses the secondary structure in water is a valid result, but it doesn't prove anything. **b)** If the authors could take the coiled structure from water and show it forms a helix on the membrane, that would be interesting. But that might not be achievable in a reasonable timeframe.

a) Reply: In classical MD simulations (without any additional bias), it is difficult to observe the transition between the folded state and unfolded state of the peptide, which is usually separated by a huge free energy barrier. To better differentiate the folding stability of the homocationic **PBLK(20)** in water *versus* near bacterial membrane, we have now studied the folding free energy profile in the different environments by 200 ns enhanced sampling using meta-dynamics. The number of intramolecular H-bonds was selected as the collective variable. We observed that the helical structure on membrane is separated from the unfolded state by an energy barrier larger than 30 kJ/mol, thus guaranteeing its stability and its resistance to unfolding. In aqueous environment, on the other hand, the free energy profile shows a much lower barrier. As a consequence, the helical structure in water cannot be maintained stably. Our results prove that the folding free energy profile is greatly modified by the anionic membrane environment, leading to the stable helical conformation of the PBLK on the membrane.

The detailed parameters have been updated in Supplementary Information Section 2.8 (page S44) and data are presented as Figure S12f and g (as below & on page S47):

Figure S12 (f) The folding free energy profiles of **PBLK(20)** in different environments. The folded state is on the right side with larger number of intramolecular H-bonds. In the membrane environment, the folded state is separated by a large free energy barrier (33 kJ/mol) from the unfolded state; in contrast, a relatively small barrier (11 kJ/mol) is found in the water environment.

Figure S12 (g) The transition from the folded state to the unfolded state in the membrane environment is more difficult than in water environment because of the higher free energy barrier.

b) Reply: We also tried “folding” simulations with the two sequences (seq1 and seq2 mentioned in Figure S12b), the initial structures are in a random coil form. Unfortunately, within 500 ns simulation time, the random coil to helix transition did not occur.

23) Fig 3e proposes quite specific structural interactions with no evidence. In particular, (II) is quite explicit about the assembly of the lipids and the helix – this is completely speculative and together with (III) and (IV) should be removed.

Reply: Figure 3e has been removed.

The Discussion requires significant revision:

24) Page 12, line 277. “The reduced tendency of the block copolymer to bind mammalian membranes is probably linked to their less negatively charged surface”. As indicated above, structural change is not indicative of binding strength, this needs to be restated.

Reply: The statement that correlates binding strength to conformation change has been deleted and the sentence has been revised in the main text to the following:

(new page 13, line 306-307)

“The reduced tendency of the block copolymer to bind mammalian membranes is probably linked to their less negatively charged surface, ~~which does not trigger the conformation change of the cationic block to its more membrane active radially amphiphilic helix conformation.~~”

25) Page 12, line 286. “expose a sheath of cationic charges and a regular helical main chain hydrophobic inner core.” This sentence is nonsensical. Firstly, what is a “sheath” and secondly the hydrophobic core is not exposed.

Reply: Sorry for the unhelpful wordings. The new sentence has been revised in main text to the following:

(new page 13, line 311-312):

“Upon surface-contact with bacterial membrane, the cationic block undergoes transition from a random coil in free solution to a helix to expose cationic charges.”

Old page 12, line 286:

“Upon surface-contact with bacterial membrane, the cationic block undergoes ~~activation of radial amphiphilicity (RA), i.e. it~~ transitions from a random coil in free solution to a helix to expose ~~a sheath of~~ cationic charges ~~and a regular helical main chain hydrophobic inner core.”~~”

26) Page 12, line 287. “Computer simulation shows that the helix probably contributes towards strong interactions with the anionic bacterial lipids”. The simulation showed no evidence of strong interactions.

Reply: The sentence has been removed.

27) Page 13, line 293. “Our block copolymer possesses unique bacteria-triggered radial amphiphilicity (RA) in which the hydrophobic domain (backbone) is hidden in free solution to reduce hemolysis and toxicity, but the cationic charges are selectively regularized and concentrated whilst the hydrophobic backbone is more exposed by anionic bacterial lipids to then result in strong bacteria-copolymer interactions.”

I am not enamoured with the term “radial amphiphilicity”. However, peptide backbones are not considered the hydrophobic domain in amphipathic sequences. Even though the authors consider the beta-peptide backbone to be hydrophobic, once helical structure is adopted, the side chains dominate the interactive topology. Also, the term “selectively regularised and concentrated” makes no sense – the whole sentence needs to be revised.

Reply: The sentences have been revised in the manuscript to the following:

New page 13, line 312-318:

“The block copolymer possesses a unique bacteria-triggered surfactant effect – the cationic block adsorbs onto the negatively charged bacterial envelope while the hydrophilic sugar block has a strong tendency to promote dissolution, resulting in a “surfactant-like” solvation of bacteria from biofilm. The amine group of the cationic block dominates the interactive topology with erythrocytes but its hydrophilicity minimizes hemolysis. Further, the neutral sugar block also increases the hydrophilicity of the block copolymer.”

Old page 13, line 293-302:

“Our block copolymer possesses unique bacteria-triggered ~~radial amphiphilicity (RA) in which the hydrophobic domain (backbone) is hidden in free solution to reduce hemolysis and toxicity, but the cationic charges are selectively regularized and concentrated whilst the hydrophobic backbone is more exposed by anionic bacterial lipids to then result in strong bacteria-copolymer interactions.~~ The bacteria-triggered RA in the cationic block results in bacteria induced surfactant effect in the block copolymer molecules that are otherwise hydrophilic and non-mammalian cell binding in free solution, which contrasts with the inherent amphiphilicity of other reported antimicrobials such as classical surfactants⁴⁷, thereby limiting hemolytic activity.”

28) Page 13, line 316-317. “The block copolymer also appears superior to commercial non-bactericidal biofilm dispersal agents such as dispersin B or DNase. The authors have not presented any data to substantiate this claim – delete.

Reply: The statement related to dispersin B and DNase has been deleted in the manuscript.

In summary, while the peptide has interesting selective antibacterial properties, the authors have attempted to provide a structural mechanism for this activity but have over-interpreted experimental data leading to unsubstantiated conclusions.

Reply: Thank you for your encouragement that our peptide has interesting selective antibacterial properties. We have now rewritten/removed some parts about the conformation change and amphiphilicity (Q13, Q24-Q28) as suggested by your kind and helpful comments.

- 1 Zapotoczna, M., O'Neill, E. & O'Gara, J. P. Untangling the Diverse and Redundant Mechanisms of *Staphylococcus aureus* Biofilm Formation. *PLOS Pathogens* **12**, e1005671, doi:10.1371/journal.ppat.1005671 (2016).
- 2 Roch, M. *et al.* Daptomycin resistance in clinical MRSA strains is associated with a high biological fitness cost. *Frontiers in microbiology* **8**, 2303 (2017).
- 3 Gopal, R. *et al.* Synergistic effects and antibiofilm properties of chimeric peptides against multidrug-resistant *Acinetobacter baumannii* strains. *Antimicrobial agents and chemotherapy* **58**, 1622-1629 (2014).

REVIEWERS' COMMENTS:

Reviewer #1 (Remarks to the Author):

The authors have made substantial revisions to the paper and have answered satisfactorily each of the major points that I raised in my report on the original document. In doing so they have introduced new data into the main body of the paper and the supplementary data. They have expanded the range of MRSA strains tested for susceptibility including those with resistance to vancomycin and daptomycin, they have shown that the peptide dispersed biofilm formed by the two distinct mechanisms exhibited by different types of staphylococci, they have shown that the peptide retains activity against daptomycin resistant *S.aureus*, they have examined possible synergy with beta lactams, they have expanded the Gram positive bacteria tested and finally they have shown that resistance does not emerge following prolonged passage in sub-MIC concentrations in comparison to ciprofloxacin.

There are a couple of minor points concerning presentation

Line 43 introduce the definite article "the bacterial envelope"

Line 49 change to "subpopulations which are antibiotic-tolerant due to metabolic inactivity"

Line 81 change to "maintaining a good bactericidal effect"

Materials and methods should contain a section "Bacterial strains" if only to point out that details of the provenance and sources of strains are in the supplementary section

Line 234 change to "phenotypically drug resistant"

Line 297 I think the drug kills NON-replicating, antibiotic tolerant persisters

Reviewer #2 (Remarks to the Author):

The revised manuscript is still somewhat difficult to read but it has been improved by addressing many of the reviewer comments. However, analysis and interpretation of the computer simulation and CD data is still problematic.

1. The additional CD experiments performed at a range of P:L ratios should be included in the Supp Info – as this is important to demonstrate that the appropriate conditions are used across other experiments.
2. Pages 8-9: The authors first present the simulation data and then follow up with the CD data to "corroborate" the simulation data. This is completely the wrong way around – simulation can only be validated by correlation with experimental data. The authors need to switch the order and present the CD experiments first and then followed up by simulation that suggests the possible helical structures.
3. Given the detailed methods for the computer simulation, the authors should also refer to Christofferson et al, ACS Nano, 12 (9), 2018, 9101-9109. Identifying the Coiled-Coil Triple Helix Structure of β -Peptide Nanofibers at Atomic Resolution".
4. The CD technique is "spectropolarimetry" not "spectroscopy". Please correct and "CD" can be used as the abbreviation following the initial definition of the term.
5. The description of the CD data needs to be further revised. The authors state: "Circular dichroism (CD) spectroscopy corroborates that in free solution the block co-beta peptide adopts a helix-coil conformation attributed respectively to the sugar52 and cationic53 blocks (Fig. 3c). However, in the presence of model vesicles containing anionic bacterial lipids, CD spectroscopy shows that the cationic block of the copolymer, like the cationic homopolymer

(PBLK(20)), undergoes a transition to a left-handed helix structure⁵⁴ (Fig. 3d, Supplementary Fig. S12l-n)."

The CD spectra of PDGu(20), PBLK(20) and PDGu(7)-b-PBLK(13) show evidence of secondary structure which is LIKELY to be helical. The authors should continue to be circumspect about the precise helical structure.

Furthermore as summarised below, the presence of minima at different wavelengths suggests different conformations depending on conditions and composition. However, the modelling did not reveal any differences in helical structure.

- PDGu(20) adopts structure in water, lipids and different pHs with superimposable spectra and a minimum at 220nm.
- PBLK(20) only exhibits structure at pH10 (minimum ~208nm) and in POPG (min ~214nm) (presumably under conditions which neutralises the positively-charged lysines)
- PDGu(7)-b-PBLK(13) adopts structure in all conditions, reflecting the combined contribution of PDGu(20) + PBLK(20), with the presence of defined minima under different conditions.

6. It is inconsistent to describe a polymer as adopting a non-structured random coil since there is no structure "adopted". It is now more acceptable to describe "random coil" as existing as an extended structure.

7. At the anionic bacterial lipid surface, the positive charges in the PBLK block are neutralized by anionic bacterial lipids so that lysine side chain charge-charge repulsion causing the distortion of the helical conformation is substantially reduced and the copolymer transitions to a helix-helix structure (Fig. 3b, Supplementary Fig. S12e to g). The resulting helix-helix structure of the sugar-cationic block copolymer binds well to the anionic bacterial membrane (Supplementary Fig. S12h to k).

a. What segment do the authors refer to here – is the non-helical lysine segment distorting the helical PD Gu segment ? Please revise to clarify.

b. Remove "well" as there is no evidence of strength of binding.

8. Page 13: The authors state: "Upon surface-contact with bacterial membrane, the cationic block undergoes transition from a random coil in free solution to a helix to expose cationic charges." However, the cationic charges are exposed in ALL conformations, the authors need to modify this description.

9. Page 13: The authors state: "The block copolymer possesses a unique bacteria-triggered surfactant effect – the cationic block adsorbs onto the negatively charged bacterial envelope while the hydrophilic sugar block has a strong tendency to promote dissolution, resulting in a "surfactant-like" solvation of bacteria from biofilm." However, the data shown in Fig 2f shows that PBLK(20) causes dissolution of the bacterial cell wall. Can the authors clarify this apparent contradiction and revise this mechanistic description.

Reply to Reviewers

Please find our point-by-point responses below, where reviewers' comments are in black, our responses are in **red**, and the changes made in the manuscript are in **blue**. A few minor changes made by ourselves are marked in **green**.

Reviewers' comments:

Reviewer #1 (Remarks to the Author):

The authors have made substantial revisions to the paper and have answered satisfactorily each of the major points that I raised in my report on the original document. In doing so they have introduced new data into the main body of the paper and the supplementary data. They have expanded the range of MRSA strains tested for susceptibility including those with resistance to vancomycin and daptomycin, they have shown that the peptide dispersed biofilm formed by the two distinct mechanisms exhibited by different types of *staphylococci*, they have shown that the peptide retains activity against daptomycin resistant *S. aureus*, they have examined possible synergy with beta lactams, they have expanded the Gram-positive bacteria tested and finally they have shown that resistance does not emerge following prolonged passage in sub-MIC concentrations in comparison to ciprofloxacin.

Reply: Thank you very much for your kind recommendations and helpful comments.

There are a couple of minor points concerning presentation:

1. Line 43 introduce the definite article “the bacterial envelope”

Reply: The sentence in old line 43 has been revised to the following:

(new page 2, line 43-44)

“The copolymer displays bacteria-activated surfactant-like properties, resulting from contact with **the** bacterial envelope.”

2. Line 49 change to “subpopulations which are antibiotic-tolerant due to metabolic inactivity”

Reply: The sentence in old line 49 has been revised to the following:

(new page 3, line 50-51)

“Compounding the difficulty of treating antibiotic resistant strains is the presence of persisters, **a**-subpopulations which **is** **are** antibiotics-tolerant due to **its** metabolic inactivity”

3. Line 81 change to “maintaining a good bactericidal effect”

Reply: The sentence in old line 81 has been revised to the following:

(new page 4, line 82-83)

“.....to reduce hemolysis whilst maintaining a good bactericidal effect⁴⁴⁻⁴⁷.”

4. Materials and methods should contain a section “Bacterial strains” if only to point out that details of the provenance and sources of strains are in the supplementary section.

Reply: The origins of bacterial strains are now added in a sub-section “Bacterial strains” of the Methods section of the main text (new page 17, line 406-414.

5. Line 234 change to “phenotypically drug resistant”

Reply: The sentence has been revised to the following:

(new page 10, line 241-242)

“Consistent with published literature¹, non-replicating *S. aureus* was phenotypically drug resistant to antibiotics from various categories”

6. Line 297 I think the drug kills NON-replicating , antibiotic tolerant persisters

Reply: The sentence has been revised to the following:

(new page 13, line 304-305)

“**PDGu(7)-b-PBLK(13)** kills non-replicating, antibiotic-tolerant persisters, and biofilm-associated MRSA, both *in vitro* and *in vivo*.”

Reviewer #2 (Remarks to the Author):

The revised manuscript is still somewhat difficult to read but it has been improved by addressing many of the reviewer comments. However, analysis and interpretation of the computer simulation and CD data is still problematic.

Reply: Thank you very much for your helpful comments to improve our manuscript. We have now rewritten/removed some parts as suggested.

1. The additional CD experiments performed at a range of P:L ratios should be included in the Supp Info – as this is important to demonstrate that the appropriate conditions are used across other experiments.

Reply: The CD data with different P:L ratios are now added as new Supplementary Fig. 33 d to f in page S28 (also shown below).

Supplementary Figure 33. (d to f) Molar ellipticity $[\theta]$ circular dichroism spectra in the presence of anionic POPG liposome at different P:L ratios. (d) PDGu(20), (e) PBLK(20) and (f) PDGu(7)-b-PBLK(13).

The following sentences have been added in the Supplementary Information:

(new page S38):

“To verify that the appropriate peptide:lipid (P:L) conditions were used, we further conducted the CD measurements at low (5:1) to high (around 1:80) P:L ratios while keeping the peptide concentration constant (Supplementary Figure 33d-f). The CD signal increases as the P:L ratio increases, but it plateaus at the charge neutralization ratio and beyond; the charge neutralization P:L ratios for the homocationic PBLK(20) and the copolymer PDGu(7)-b-PBLK(13) are 1:20 and 1:13 respectively. Hence our chosen P:L ratios of 1:79 and 1:89 for CD studies in Supplementary Figure 33a-c are beyond the saturation ratio for both charged polymers.”

2. Pages 8-9: The authors first present the simulation data and then follow up with the CD data to “corroborate” the simulation data. This is completely the wrong way around – simulation can only be validated by correlation with experimental data. The authors need to switch the order and present the CD experiments first and then followed up by simulation that suggests the possible helical structures.

Reply: As suggested, we have rearranged the data to present CD experiment result first and then computer simulation (main text page 8, line 197-213).

3. Given the detailed methods for the computer simulation, the authors should also

refer to Christofferson et al, ACS Nano, 12 (9), 2018, 9101-9109. Identifying the Coiled-Coil Triple Helix Structure of β -Peptide Nanofibers at Atomic Resolution”.

Reply: The abovementioned reference has been added as new ref 24 in the Supplementary Information.

(new page S41)

“As reported, beta-peptides could form different helical structures including 14 helix, 12 helix, 10 helix and 8 helix^{23,24}, which are named based on the number of atoms comprising the H-bond ring.”

Ref. 24: Christofferson, A. J. *et al.* Identifying the Coiled-Coil Triple Helix Structure of beta-Peptide Nanofibers at Atomic Resolution. *ACS Nano* **12**, 9101-9109, doi:10.1021/acsnano.8b03131 (2018).

4. The CD technique is “spectropolarimetry” not “spectroscopy”. Please correct and “CD” can be used as the abbreviation following the initial definition of the term.

Reply: The “spectroscopy” has been corrected to “spectropolarimetry” in the main text, and “CD” is now used in the subsequent contents after the initial definition.

(new page 8, line 197-199)

“Circular dichroism (CD) ~~spectroscopy~~ spectropolarimetry showed that in free solution the block co-beta-peptide adopts a helix-coil conformation attributed respectively to the sugar⁵² and cationic⁵³ blocks (Fig. 3a).”

5. The description of the CD data needs to be further revised. The authors state: “Circular dichroism (CD) spectroscopy corroborates that in free solution the block co-beta peptide adopts a helix-coil conformation attributed respectively to the sugar⁵² and cationic⁵³ blocks (Fig. 3c). However, in the presence of model vesicles containing anionic bacterial lipids, CD spectroscopy shows that the cationic block of the copolymer, like the cationic homopolymer (**PBLK(20)**), undergoes a transition to a left-handed helix structure⁵⁴ (Fig. 3d, Supplementary Fig. S12l-n).”

(a) The CD spectra of **PDGu(20)**, **PBLK(20)** and **PDGu(7)-b-PBLK(13)** show evidence of secondary structure which is **LIKELY** to be helical. The authors should continue to be circumspect about the precise helical structure.

Reply: We have added “likely” in the sentence as suggested to be circumspect about the precise helical structure.

(new page 8, line 197-202)

“Circular dichroism (CD) spectropolarimetry showed that in free solution the block co-beta-peptide **likely** adopts a helix-coil conformation attributed respectively to the sugar⁵² and cationic⁵³ blocks (Fig. 3a). However, in the presence of model vesicles containing anionic bacterial lipids, CD spectrum shows that the cationic block of the copolymer, like the cationic homopolymer (**PBLK(20)**), undergoes a transition to **likely** a left-handed helix structure⁵⁴ (Fig. 3b, Supplementary Fig. 33a-f).”

(b) Furthermore as summarised below, the presence of minima at different

wavelengths suggests different conformations depending on conditions and composition. However, the modelling did not reveal any differences in helical structure.

- **PDGu(20)** adopts structure in water, lipids and different pHs with superimposable spectra and a minimum at 220nm.
- **PBLK(20)** only exhibits structure at pH10 (minimum ~208nm) and in POPG (min ~214nm) (presumably under conditions which neutralises the positively-charged lysines)
- **PDGu(7)-b-PBLK(13)** adopts structure in all conditions, reflecting the combined contribution of **PDGu(20)** + **PBLK(20)**, with the presence of defined minima under different conditions.

Reply: Our modelling actually did show difference in the helical structure of the **PDGu** and **PBLK**, and we have added the following sentences in the manuscript:

(main text new page 9, line 217-219)

“Our computer simulation corroborated that the **PDGu** and **PBLK** blocks adopt different helical conformations, with 3.5 residues/turn and 3 residues/turn respectively (Supplementary Fig. 34j, Supplementary Note).”

(Supplementary Information new page S43)

“Although these two helices are both left-handed, they are structurally different from each other: one rise in the **PBLK** helix contains 3 residues with a height of 0.476 ± 0.014 nm, while that in the **PDGu** helix has 3.5 residues with a height of 0.677 ± 0.021 nm.”

Supplementary Figure 34. (j) H-bond formation in the border between the **PDGu** and the **PBLK** blocks of the **PDGu(7)-b-PBLK(13)**. The height and number of residues per helical turn are different in the two helices. The copolymer backbone is shown as stick model while the rest of molecule is shown as line model.

6. It is inconsistent to describe a polymer as adopting a non-structured random coil since there is no structure “adopted”. It is now more acceptable to describe “random coil” as existing as an extended structure.

Reply: The sentence has been revised in the main text to the following:

(new page 9, line 205-207)

“In free solution, electrostatic repulsion between the lysine side chains of the cationic block causes the cationic block to ~~adopt~~ exist as a random coil conformation”

7. “At the anionic bacterial lipid surface, the positive charges in the **PBLK** block are neutralized by anionic bacterial lipids so that lysine side chain charge-charge repulsion causing the distortion of the helical conformation is substantially reduced and the copolymer transitions to a helix-helix structure (Fig. 3b, Supplementary Fig. S12e to g). The resulting helix-helix structure of the sugar-cationic block copolymer binds well to the anionic bacterial membrane (Supplementary Fig. S12h to k).”

(a) What segment do the authors refer to here – is the non-helical lysine segment distorting the helical **PDGu** segment? Please revise to clarify.

Reply: Sorry for the confusion. Please see the clarified rewrite of these sentences now below:

(new page 9, line 207-211)

“At the anionic bacterial lipid surface, the positive charges in the **PBLK** block of the copolymer are neutralized by anionic bacterial lipids so that the lysine side chain charge-charge repulsion causing the distortion of the helical conformation of the copolymer **PBLK** block is substantially reduced and the copolymer transitions from a helix-coil structure to a helix-helix structure”

(b) Remove “well” as there is no evidence of strength of binding.

Reply: The word “well” has been removed from the following sentence:

(new page 9, line 212-213)

“The resulting helix-helix structure of the sugar-cationic block copolymer binds ~~well~~ to the anionic bacterial membrane”

8. Page 13: The authors state: “Upon surface-contact with bacterial membrane, the cationic block undergoes transition from a random coil in free solution to a helix to expose cationic charges.” However, the cationic charges are exposed in ALL conformations, the authors need to modify this description.

Reply: The sentence has been revised to the following:

(new page 13, line 318-319)

“Upon surface-contact with bacterial membrane, the cationic block undergoes transition from a random coil in free solution to a helix ~~to expose cationic charges~~”

9. Page 13: The authors state: “The block copolymer possesses a unique bacteria-triggered surfactant effect – the cationic block adsorbs onto the negatively charged

bacterial envelope while the hydrophilic sugar block has a strong tendency to promote dissolution, resulting in a “surfactant-like” solvation of bacteria from biofilm.” However, the data shown in Fig 2f shows that **PBLK(20)** causes dissolution of the bacterial cell wall. Can the authors clarify this apparent contradiction and revise this mechanistic description.

Reply: Sorry for the confusion. We have rewritten the sentences to clarify this point in the main text:

(new page 13, line 319-326)

“The block copolymer possesses a unique bacteria-triggered surfactant effect that contributes to biofilm dispersal – the cationic block adsorbs onto the negatively charged bacterial envelope while the hydrophilic sugar block has a strong tendency to promote dissolution, resulting in a “surfactant-like” solvation of bacteria from biofilm. The block copolymer forms an anti-fouling **PDGu** layer around the bacteria. Conversely, the homocationic **PBLK(20)** led to pore formation (Fig. 2b, c, f), like other AMPs³⁶, but without promoting biofilm detachment (Fig. 4h); this is probably linked to the inability of the homocationic polymer to form an anti-fouling layer around the bacteria.”